# VIPO: Value Function Inconsistency Penalized Offline Reinforcement Learning

## Abstract

Offline reinforcement learning (RL) learns effective policies from pre-collected datasets, offering a practical solution for applications where online interactions are risky or costly. Model-based approaches are particularly advantageous for offline RL, owing to their data efficiency and generalizability. However, due to inherent model errors, model-based methods often artificially introduce conservatism guided by heuristic uncertainty estimation, which can be unreliable. In this paper, we introduce VIPO, a novel model-based offline RL algorithm that incorporates self-supervised feedback from value estimation to enhance model training. Specifically, the model is learned by additionally minimizing the inconsistency between the value learned directly from the offline data and the one estimated from the model. We perform comprehensive evaluations from multiple perspectives to show that VIPO can learn a highly accurate model efficiently and consistently outperform existing methods. In particular, it achieves state-of-the-art performance on almost all tasks in both D4RL and NeoRL benchmarks. Overall, VIPO offers a general framework that can be readily integrated into existing model-based offline RL algorithms to systematically enhance model accuracy. Our code is available at `https://anonymous.4open.science/r/vipo2025-8FD4`.

## 1 Introduction

Offline reinforcement learning (RL) (Lange et al., 2012; Levine et al., 2020) aims to learn effective policies exclusively from a pre-collected behavior dataset, eliminating the need for online interaction with the environment. This approach offers a compelling solution for tasks where online interactions entail significant risks or exorbitant costs. Due to its potential to transform static datasets into powerful decision-making systems, offline RL has gained popularity in recent years (Kalashnikov et al., 2018; Prudencio et al., 2023). While *off-policy* RL algorithms can in principle be directly applied to the offline datasets, recent studies show that they can perform poorly when applied offline (Fujimoto et al., 2019; Kumar et al., 2019). This is primarily attributed to the distribution shift between the learned and the behavior policies, as learning policies beyond the behavior policy requires querying the actions not observed in the dataset. Similarly, value function evaluation on out-of-distribution (OOD) actions is usually inaccurate. In turn, maximizing an inaccurate value function during policy improvement often overestimates the value of OOD actions, which is a core challenge in offline RL.

To mitigate overestimation, a common paradigm in offline RL is to incorporate conservatism into algorithm design. Model-free offline RL algorithms (Fujimoto et al., 2018; Wu et al., 2019; Kumar et al., 2020; Wang et al., 2022) achieve this by learning a pessimistic value function (value-based approach) to discourage choosing OOD actions or constraining the learned policy (policy-based approach) to avoid visiting such actions. However, these algorithms often suffer from overly conservatism as many of them solely learn from the dataset (Wang et al., 2018; Chen et al., 2020; Kostrikov et al., 2021). Meanwhile, it is crucial to balance conservatism and generalization since being overly conservative hinders finding a better policy. In contrast, model-based offline RL algorithms (Kidambi et al., 2020; Yu et al., 2020; Sun et al., 2023) ensure conservative by learning a pessimistic dynamics model which penalizes the value of OOD actions. Additionally, model-based algorithms have the potential for broader generalization as they can incorporate dynamics models to generate synthetic data that are not present in the dataset (Yu et al., 2020). This advantage makes model-based methods particularly well-suited for offline RL, motivating their further investigation in this paper.

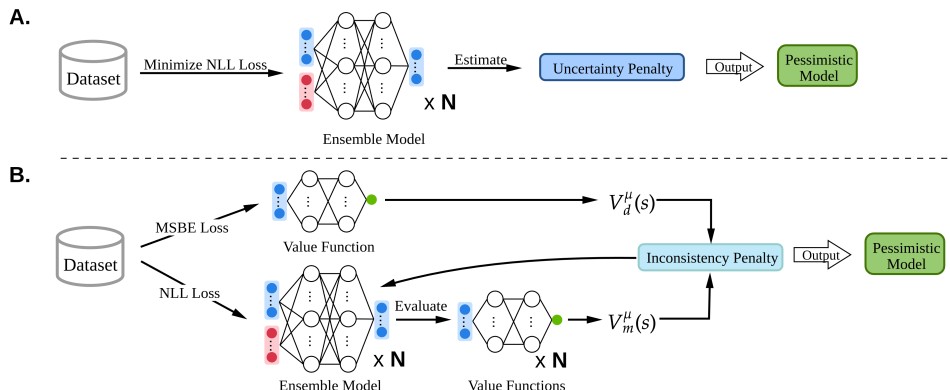

Figure 1: Comparison of VIPO with previous model-based approaches for learning a pessimistic dynamics model. **(A)** Previous model-based methods make a single use of data to learn an ensemble of models and then use its uncertainty to apply *ad-hoc*, pessimistic adjustments to the model predictions. **(B)** VIPO leverages the data in two ways: (1) it learns a value function $V_d^\mu(s)$ directly by minimizing the mean-square Bellman error (MSBE) loss; (2) it first learns the dynamics model by minimizing the negative log-likelihood (NLL) loss and then estimate the ensemble value functions $V_m^\mu(s)$. VIPO utilizes the discrepancy between these two types of value functions as an additional self-supervised loss to improve the model learning performance.

Most prior model-based algorithms rely on uncertainty estimation to incorporate conservatism. They commonly train an ensemble of $N$ models through maximum likelihood on the dataset and quantify uncertainty based on the standard deviation among the predictions of these $N$ models. Specifically, (Yu et al., 2020) employs the max-aleatoric uncertainty quantifier, (Kidambi et al., 2020) adopts the max-pairwise-diff uncertainty quantifier, and (Lu et al., 2021) utilizes the ensemble-standard-deviation uncertainty quantifier. Nevertheless, it was shown that the uncertainties of their learned models using these methods are inaccurate and unreliable when dealing with complex datasets in practice (Ovadia et al., 2019). This aligns with our findings: as shown empirically in Section 4.2, even in Gym tasks the uncertainties estimated by MOPO-learned models fail to reflect the expected increase as the data-drop ratio grows, whereas those from VIPO-learned models track this trend well. This indicates that VIPO captures epistemic uncertainty due to data loss more effectively than MOPO and thus is likely to learn a more accurate model.

In this paper, we introduce **V**alue Function **I**nconsistency **P**enalized **O**ffline Reinforcement Learning (VIPO), a model-based offline RL algorithm that aims to learn a highly accurate model by integrating a value function inconsistency loss into model training. VIPO differs fundamentally from prior methods in two key aspects. First, our approach makes a dual-usage of the data in a self-supervised fashion to improve the model training accuracy. As illustrated in Figure 1, previous model-based RL methods make only a single-purpose use of the data: they learn an ensemble of models and obtain a pessimistic model by applying ad-hoc adjustments to the model outputs based on the uncertainty estimated from the ensemble. In contrast, VIPO leverages the data in two complementary ways. The first use of the data is to directly learn an approximated value function of the behavior policy by minimizing the empirical Mean Square Bellman Error (MSBE). The second use of the data is to learn a model by minimizing the negative log-likelihood loss (NLL), followed by value evaluation to obtain another approximated value function of the behavior policy. If the learned model is accurate, the two value functions should coincide owing to the *uniqueness* of the value function, and their alignment (or lack thereof) provides a direct quantifier of model accuracy. Since offline RL must rely entirely on the fixed dataset, effectively exploiting it is critical. Compared with prior approaches that use the data for a single purpose, our method leverages it in two complementary ways.

Second, unlike previous works which do not mitigate the ensemble-model uncertainty in the training stage, we proactively incorporate the value function inconsistency into the model training loss in a self-supervised manner to improve model learning. By establishing the analytical expression for the gradient of the augmented loss in Theorem 3.3, we enable the model to be trained via gradient descent, ensuring a balance between data fidelity and alignment in value functions. To the best of our knowledge, no existing model-based offline RL method has incorporated either uncertainty or inconsistency into the training stage. Our approach is the first to proactively mitigate inconsistency during training, instead of ad-hoc adjustments of the model predictions.

Our empirical results demonstrate from multiple perspectives that incorporating value function inconsistency into model error enhances model accuracy, highlighting the effectiveness of the aforementioned two novel designs. VIPO offers a general framework that can be readily integrated into existing model-based offline RL algorithms to systematically enhance model accuracy. We achieve this without relying on expensive architectures such as diffusion or transformer, rather than by comprehensive exploitation of data. Moreover, VIPO outperforms prior offline RL algorithms and achieves state-of-the-art (SOTA) performance on most tasks of both D4RL and NeoRL benchmarks.

## 2 PRELIMINARIES

### 2.1 OFFLINE REINFORCEMENT LEARNING

RL problems are commonly formulated within the framework of a Markov Decision Process (MDP), defined by the tuple $\mathcal{M} = (\mathcal{S}, \mathcal{A}, r, \rho_0, P, \gamma)$. Here, $\mathcal{S}$ denotes the state space, $\mathcal{A}$ represents the action space, $r(s, a) : \mathcal{S} \times \mathcal{A} \to [-r_{\max}, r_{\max}]$ is a bounded reward function, $\rho_0(s)$ specifies the initial state distribution, $P(s' \mid s, a) : \mathcal{S} \times \mathcal{A} \times \mathcal{S} \to \mathbb{R}_+$ defines the transition kernel, and $\gamma \in [0, 1)$ is the discount factor. Given a policy $\pi(\cdot \mid s)$, its performance can be evaluated by the expected cumulative long-term reward, defined as

$$J(\pi) = \mathbb{E}_{s_0 \sim \rho_0, a_t \sim \pi, s_{t+1} \sim P} \left[ \sum_{t=0}^{\infty} \gamma^t r(s_t, a_t) \right]. \tag{1}$$

The goal of RL is to learn a policy $\pi(\cdot \mid s)$ that maximizes the expected cumulative long-term reward, i.e., $\pi^* = \arg\max_\pi J(\pi)$.

As shown in Eq. (1), the classical RL framework requires online interactions with the environment $P$ during training. In contrast, the offline RL learns only from a fixed dataset $\mathcal{D} = \{(s, a, r, s')\}$ collected by the behavior policy $\mu(\cdot \mid s)$, where $s$, $a$, $r$, and $s'$ denote the state, action, reward, and next state, respectively. That is, it aims to find the best possible policy solely from $\mathcal{D}$ without additional interactions with the environment.

### 2.2 MODEL-BASED OFFLINE RL ALGORITHMS

Model-based offline RL algorithms aim to derive the optimal policy by utilizing a learned dynamics model. These methods define a pessimistic MDP $\widehat{\mathcal{M}} = \langle \mathcal{S}, \mathcal{A}, \hat{r}, \rho_0, \hat{P}, \gamma \rangle$, which shares the same state and action spaces as the original MDP but employs the learned transition dynamics $\hat{P}(s' \mid s, a)$ and reward function $\hat{r}(s, a)$. $\hat{P}$ and $\hat{r}$ are typically estimated through supervised regression on the dataset $\mathcal{D}$, with additional incorporation of uncertainty penalties to discourage exploration of out-of-distribution (OOD) actions. These pessimistic models then serve as proxies for the real environment, enabling the simulation of transitions and subsequent use for planning.

The success of model-based RL algorithms hinges on the ability to learn an accurate model. Once the model is learned, various approaches, such as model predictive control (MPC) (Williams et al., 2017), dynamic programming (Munos & Szepesvári, 2008), or model-based policy optimization (Janner et al., 2019), can be employed to recover the optimal policy by solving the pessimistic MDP $\widehat{\mathcal{M}}$.

## 3 OUR METHOD

### 3.1 VALUE FUNCTION LEARNING

We first learn an approximated value function of the behavior policy directly from the dataset. In the MDP $\mathcal{M} = (\mathcal{S}, \mathcal{A}, r, \rho_0, P, \gamma)$, under the behavior policy $\mu(\cdot \mid s)$, the Bellman backup for obtaining the corresponding value function $V$ is defined as

$$\mathcal{T}^\mu V(s) := \mathbb{E}_{a \sim \mu(\cdot \mid s), s' \sim P(\cdot \mid s, a)} \left[ r(s, a) + \gamma V(s') \right].$$

Therefore, from the offline dataset $\mathcal{D}$, we can define the following empirical Bellman operator:

$$\widehat{\mathcal{T}}_d^\mu V(s) := \begin{cases} \dfrac{1}{|\mathcal{D}_s|} \displaystyle\sum_{(s,a,r,s') \in \mathcal{D}_s} \left[ r + \gamma V(s') \right], & \text{if } |\mathcal{D}_s| > 0, \\ 0, & \text{otherwise,} \end{cases} \tag{2}$$

where $\mathcal{D}_s \subseteq \mathcal{D}$ is the set of transitions in the dataset $\mathcal{D}$ that begin in state $s$, $|\mathcal{D}_s|$ is the number of such transitions, and $\gamma$ is the discount factor.

We then show that the empirical Bellman operator has a unique fixed point.

**Proposition 3.1.** *The empirical Bellman operator $\widehat{\mathcal{T}}_d^\mu$ has a unique fixed point $V_d^\mu(s)$ such that*

$$\widehat{\mathcal{T}}_d^\mu V_d^\mu(s) = V_d^\mu(s). \tag{3}$$

From Theorem 3.1, we can learn a value function $V_d^\mu(s)$ from the offline dataset $\mathcal{D}$. Intuitively, when $\mathcal{D}$ is densely sampled from rollouts of the behavior policy $\mu$, the learned $V_d^\mu(s)$ provides an accurate approximation of the true value function $V^\mu(s)$, such that $V_d^\mu(s) \approx V^\mu(s)$.

We then train another approximated value function corresponding to the behavior policy with the help of a learned model. We learn an approximate dynamics model $P_\theta(s', r \mid s, a)$ through maximum likelihood estimation and define the Bellman operator associated with $P_\theta$ as follows:

$$\widehat{\mathcal{T}}_m^\mu V(s) := \mathbb{E}_{a \sim \mu(\cdot \mid s), (r, s') \sim P_\theta}\left[r + \gamma V(s')\right]. \tag{4}$$

Note that if the learned model $P_\theta$ accurately approximates the true model ($P$ and $r$), it follows that $\widehat{\mathcal{T}}_m^\mu$ becomes equivalent to $\mathcal{T}^\mu$.

**Proposition 3.2.** *The Bellman operator induced by $P_\theta$, denoted $\widehat{\mathcal{T}}_m^\mu$, has a unique fixed point $V_m^\mu(s)$ such that*

$$\widehat{\mathcal{T}}_m^\mu V_m^\mu(s) = V_m^\mu(s). \tag{5}$$

From Theorem 3.2, we can iteratively apply $\widehat{\mathcal{T}}_m^\mu$ to obtain the unique value function $V_m^\mu$, representing the value function learned from the model. This value function serves as another approximation of $V^\mu(s)$, such that $V_m^\mu(s) \approx V^\mu(s)$.

The core idea is that if the learned model $P_\theta$ is accurate, the value function $V_m^\mu(s)$—obtained from the model—should align with $V_d^\mu(s)$—derived from the dataset—as both aim to approximate the true value function $V^\mu(s)$. Therefore, we define the value function inconsistency loss as follows:

$$\mathcal{L}_{vic}(\theta) = \mathbb{E}_{s \sim \rho_0}\left[\left(V_d^\mu(s) - V_m^\mu(s)\right)^2\right], \tag{6}$$

where $\rho_0$ dentoes the initial distribution, and $\mathcal{L}_{vic}$ depends on $\theta$ because $V_m^\mu(s)$ is derived from the learned model $P_\theta$.

To learn the model, we begin with minimizing the negative log-likelihood, which we refer to as the original model (learning) loss,

$$\mathcal{L}_{\mathrm{ori}}(\theta) = -\mathbb{E}_{\mathcal{D}}\left[\log P_\theta(s', r \mid s, a)\right], \tag{7}$$

where $P_\theta$ is parameterized by a neural network that models the next state and reward as a Gaussian distribution conditioned on the current state and action:

$$P_\theta(s_{t+1}, r_t \mid s_t, a_t) = \mathcal{N}(\mu_\theta(s_t, a_t), \Sigma_\theta(s_t, a_t)).$$

We incorporate the value function inconsistency loss into the original model loss to construct the following augmented model loss:

$$\mathcal{L}_{aug}(\theta) = \mathcal{L}_{ori}(\theta) + \lambda \mathcal{L}_{vic}(\theta), \tag{8}$$

where $\lambda > 0$ is a user-chosen hyperparameter that balances the contributions of the two loss components. Thus, $\mathcal{L}_{aug}$ serves as the overall loss for training the dynamics model which aims to optimize both the original model loss and the value function inconsistency loss. Note that similar to (Yu et al., 2020), to improve robustness, we learn an ensemble of $N$ dynamics models, each with different initializations.

### 3.2 MODEL GRADIENT THEOREM

To optimize $\mathcal{L}_{aug}$, it is essential to compute its gradient. However, in deriving the gradient of the augmented loss function, a significant challenge emerges from the implicit dependency of the model-learned value function $V_m^\mu(s)$ on the dynamics model parameters $\theta$. Unlike the original loss term

---

**Algorithm 1** VIPO: Value Function Inconsistency Penalized Offline Reinforcement Learning

---

1: **Require:** Offline dataset $\mathcal{D} = \{(s_i, a_i, r_i, s_i')\}_{i=1}^N$; Initial model parameters $\theta$; Regularization coefficient $\lambda$; Learning rate $\eta$; Maximum number of iterations $T$.
2: Set iteration counter $t \leftarrow 0$.
3: **while** not reaching maximum iterations $T$ or convergence criterion **do**
4:     Compute original model loss via Eq. (7).
5:     Use the offline dataset $\mathcal{D}$ to compute the value function $V_d(s)$ by minimizing Eq. (9).
6:     Use the current model $P_\theta$ to compute the model value function $V_m(s)$ by minimizing Eq. (10).
7:     Update $\theta$ using gradient descent via Eq. (11).
8:     Update the iteration counter: $t \leftarrow t + 1$.
9:     Check convergence criterion; if satisfied, exit the loop.
10: **end while**
11: Obtain the optimized model parameters $\theta^* = \theta$. Let $\widehat{\mathcal{M}}$ be the MDP with learned dynamics model $P_{\theta^*}$.
12: (OPTIONAL) Use a behavior cloning approach to estimate the behavior policy $\mu$.
13: Run any RL algorithm on $\widehat{\mathcal{M}}$ until convergence to obtain $\pi_{\text{out}} \leftarrow \text{PLANNER}(\widehat{\mathcal{M}}, \pi_{\text{int}} = \mu)$.

---

$\mathcal{L}_{ori}$, whose gradient can be readily computed through automatic differentiation, the value function $V_m^\mu(s)$ is obtained via a recursive Bellman backup defined in Eq. (4) that involves multiple steps of state transitions and discounted future rewards. This recursive structure embeds a hidden dependence on $\theta$, thereby precluding the direct application of conventional differentiation techniques.

In the following, we establish the analytical expression for the gradient of the augmented loss function, which forms the foundation for implementing VIPO. A key component of our approach is the recursive unrolling of gradients, allowing us to track the impact of model parameters across multiple transitions.

Let $\rho_\theta^\mu(s \to s', t)$ denote the state density at $s'$ after transitioning for $t$ time steps from state $s$ under the policy $\mu$ and the learned model $P_\theta$. Denote the (improperly) discounted state transition probability from $s$ to $s'$ by $d_\theta^\mu(s, s')$, defined as $d_\theta^\mu(s, s') = \sum_{t=0}^\infty \gamma^t \rho_\theta^\mu(s \to s', t)$.

We have the following Model Gradient Theorem.

**Theorem 3.3** (Model Gradient Theorem). *Let $\theta$ represent the parameters of the dynamics model $P_\theta$. Denote $V_d^\mu(\cdot)$ as the value function learned from the offline dataset $\mathcal{D}$, satisfying Eq. (3), and $V_m^\mu(\cdot)$ as the value function learned from the dynamics model $P_\theta$, satisfying Eq. (5). Then:*

$$
\begin{aligned}
\nabla_\theta \mathcal{L}_{aug}(\theta) = \nabla_\theta \mathcal{L}_{ori}(\theta) - 2\lambda \mathbb{E}_{s \sim \rho_0, s' \sim d_\theta^\mu, a' \sim \mu, (r', s'') \sim P_\theta} \big[ \big(V_d^\mu(s) \\
- V_m^\mu(s)\big)\big(r' + \gamma V_m^\mu(s'')\big) \nabla_\theta \log P_\theta(s'', r'|s', a') \big]
\end{aligned}
$$

Theorem 3.3 provides the gradient of the augmented model loss, which underlies the practical implementation of VIPO. The proof of Theorem 3.3 is given in Appendix B.

### 3.3 PRACTICAL ALGORITHM

We now introduce our practical algorithm, VIPO.

In VIPO, neural networks are employed to approximate the value function. To approximate $V_d^\mu(s)$, we learn $V_d(s)$ by minimizing the following empirical mean squared Bellman error (MSBE):

$$
\mathcal{L}_{V_d}(\varphi_d) = \mathbb{E}_{(s,a,r,s') \sim \mathcal{D}} \left[ \big(r + \gamma \bar{V}_d(s') - V_d(s)\big)^2 \right], \tag{9}
$$

where $\varphi_d$ is the parameter of primary state value network $V_d$ and $\bar{V}_d$ is the target state value network. $\bar{V}_d$ is obtained using an exponentially moving average of parameters of the state value network (soft update): $\bar{\varphi}_d \leftarrow \tau \varphi_d + (1 - \tau)\bar{\varphi}_d$, where $\tau \in [0, 1]$. Eq. (9) serves as the surrogate objective for learning the Bellman backup defined in Eq. (2).

We then learn $V_m(s)$ to approximate $V_m^\mu(s)$ by minimizing the following loss function:

$$
\mathcal{L}_{V_m}(\varphi_m) = \mathbb{E}_{(s,a) \sim \mathcal{D}, (r,s') \sim P_\theta} \left[ \big(r + \gamma \bar{V}_m(s') - V_m(s)\big)^2 \right], \tag{10}
$$

where $\varphi_m$ is the parameter of the primary network $V_m$, and $\bar{V}_m$ is the target network obtained by the exponentially moving average of parameters of $V_m$. Note that $r$ and $s'$ are sampled from the learned

dynamic model $P_\theta$ instead of from the offline dataset. Eq. (10) serves as the surrogate for learning the Bellman backup defined in Eq. (4).

From Theorem 3.3, computing the model gradient requires samples of the form $(s, a, r, s', a', r', s'')$. However, existing datasets provide only randomly sampled tuples $(s, a, r, s')$. Nevertheless, all benchmark tasks (and typically most challenging real-world applications) are continuous-control problems with very small sampling intervals (typically 0.008s–0.015s in standard Gym environments). Consequently, the state change over a single step is numerically insignificant. In practice, therefore, we leverage the available samples $(s, a, r, s')$ to closely approximate $(s', a', r', s'')$, and this approach proves highly effective. Then the gradient can be estimated as follows: $\nabla_\theta \log P_\theta$ can be obtained through automatic differentiation, $V_d^\mu$ is approximated by $V_d$, and $V_m^\mu$ is approximated by $V_m$. To enhance stability during the learning process, we replace $V_d$ and $V_m$ with the target value functions $\bar{V}_d$ and $\bar{V}_m$, resulting in the following surrogate gradient:

$$
\begin{aligned}
\nabla_\theta \mathcal{L}_{aug}^{surr}(\theta) = \nabla_\theta \mathcal{L}_{ori}(\theta) - 2\lambda \mathbb{E}_{(s,a)\sim\mathcal{D},(s',r)\sim P_\theta}\big[(\bar{V}_d(s) \\
- \bar{V}_m(s))(r + \gamma \bar{V}_m(s'))\nabla_\theta \log P_\theta(s',r|s,a)\big].
\end{aligned}
\tag{11}
$$

**Remark.** We next provide intuition for why the proposed surrogate gradient is effective by drawing an analogy to the policy gradient theorem (Sutton et al., 1999): $\nabla_\theta J(\theta) = \mathbb{E}_{s\sim\rho^{\pi_\theta}, a\sim\pi_\theta}[Q^{\pi_\theta}(s,a)\nabla_\theta \log \pi_\theta(a|s)]$. In our surrogate gradient, the term $r + \gamma \bar{V}_m(s')$ serves as an approximation of the Q-value corresponding to the learned model. Analogous to the policy gradient case, where it updates the policy $\pi_\theta$ to maximize $J(\theta)$, in our model gradient case, $(r + \gamma \bar{V}_m(s'))\nabla_\theta \log P_\theta(s',r|s,a)$ updates $P_\theta$ to improve the value function, where $\log P_\theta(s',r|s,a)$ plays a role similar to $\log \pi_\theta(a|s)$ in the policy gradient theorem. Specifically, $(r + \gamma \bar{V}_m(s'))\nabla_\theta \log P_\theta(s',r|s,a)$ is the direction that increases $V_m$. Multiplying this term by $-2\lambda(\bar{V}_d(s) - \bar{V}_m(s))$ ensures that when $\bar{V}_d(s) > \bar{V}_m(s)$, the gradient descent update in our algorithm updates the model to increase $\bar{V}_m$ towards $\bar{V}_d$, and vice versa. Consequently, this surrogate gradient promotes consistency between value functions.

We now present the complete algorithm flow in Algorithm 1, which outlines the dynamics model training process of VIPO. The process begins by computing the original model loss using Eq. (7). Next, $V_d(s)$ and $V_m(s)$ are learned by minimizing Eq. (9) and Eq. (10), respectively. With these components in place, the model parameters $\theta$ are updated using Eq. (11).

The key part of Algorithm 1 is the training of the model, since the success of model-based RL algorithms hinges on the ability to learn an accurate model. Once the model is learned, various approaches, such as model predictive control (MPC) (Williams et al., 2017), dynamic programming (Munos & Szepesvári, 2008), or model-based policy optimization (MBPO) (Janner et al., 2019), can be employed as the planner to recover the optimal policy by solving the surrogate MDP $\widehat{\mathcal{M}}$.

## 4    EXPERIMENTS

In this section, we present three distinct experiments to empirically demonstrate, from three perspectives, that VIPO consistently learns a highly accurate model compared to previous methods.

**Benchmark Results.** In Subsection 4.1, we evaluate VIPO's performance on the D4RL and NeoRL benchmarks. Specifically, we replace the models used in previous methods (MOPO and MOBILE) with the model trained by VIPO, while retaining their policy extraction components (i.e., the planner). This substitution results in significant performance improvements, enabling VIPO to achieve state-of-the-art performance across all tasks. We emphasize that the classic method MOPO and the pre-SOTA method MOBILE are selected merely as representative examples. In principle, VIPO can be integrated with any existing model-based offline RL algorithm, and such integration is expected to yield performance improvements

**Revisiting Uncertainty.** In Subsection 4.2, we design an experiment to reevaluate the uncertainty defined in MOPO (Yu et al., 2020). Specifically, we remove a portion of the Walker2d-medium-replay dataset and train models on the reduced dataset using both MOPO and VIPO. Intuitively, a higher drop ratio, indicating less available data about the environment, should result in higher model uncertainty. However, models trained with MOPO fail to capture this trend while models trained with VIPO exhibit a clear positive correlation.

Table 1: Normalized average returns on D4RL Gym tasks. The experiments are run on MuJoCo-"v2" datasets over 4 random seeds. r = random, m = medium, m-r = medium-replay, m-e = medium-expert. MOPO* indicates the results of MOPO retrained in "v2" datasets as reported by (Sun et al., 2023). The returns labeled with * in random tasks indicate values obtained from our training, as they were not reported in the original paper. We **bold** the highest mean.

| Task Name | TD3+BC | CQL | IQL | DQL | MOPO* | MOReL | COMBO | RAMBO | MOBILE | VIPO-MOPO | VIPO(Ours) |
|---|---|---|---|---|---|---|---|---|---|---|---|
| *halfcheetah-r* | 10.2 | 31.3 | 16.1* | 22.1* | 38.5 | 25.6 | 38.8 | 39.5 | 39.3 | 38.9 | **42.5 ± 0.2** |
| *hopper-r* | 11.0 | 5.3 | 10.8* | 17.5* | 31.7 | 53.6 | 17.9 | 25.4 | 31.9 | 30.2 | **33.4 ± 1.9** |
| *walker2d-r* | 1.4 | 5.4 | 7.7* | 14.1* | 7.4 | 37.3 | 7.0 | 0.0 | 17.9 | 9.5 | **20.0 ± 0.1** |
| *halfcheetah-m* | 42.8 | 46.9 | 47.4 | 51.1 | 73.0 | 42.1 | 54.2 | 77.9 | 74.6 | 76.3 | **80.0 ± 0.4** |
| *hopper-m* | 99.5 | 61.9 | 66.3 | 90.5 | 62.8 | 95.4 | 97.2 | 87.0 | 106.6 | 103.7 | **107.7 ± 1.0** |
| *walker2d-m* | 79.7 | 79.5 | 78.3 | 87.0 | 84.1 | 77.8 | 81.9 | 84.9 | 87.7 | 81.2 | **93.1 ± 1.8** |
| *halfcheetah-m-r* | 43.4 | 45.3 | 44.2 | 47.8 | 72.1 | 40.2 | 55.1 | 68.7 | 71.7 | 71.9 | **77.2 ± 0.4** |
| *hopper-m-r* | 31.4 | 86.3 | 94.7 | 101.3 | 103.5 | 93.6 | 89.5 | 99.5 | 103.9 | 103.8 | **109.6 ± 0.9** |
| *walker2d-m-r* | 25.2 | 76.8 | 73.9 | 95.5 | 85.6 | 49.8 | 56.0 | 89.2 | 89.9 | 80.5 | **98.4 ± 0.25** |
| *halfcheetah-m-e* | 97.9 | 95.0 | 86.7 | 96.8 | 90.8 | 53.3 | 90.0 | 95.4 | 108.2 | 103.4 | **110.0 ± 0.4** |
| *hopper-m-e* | 112.2 | 96.9 | 91.5 | 111.1 | 81.6 | 108.7 | 111.1 | 88.2 | 112.6 | 117.7 | **113.2 ± 0.1** |
| *walker2d-m-e* | 105.7 | 109.1 | 109.6 | 110.1 | 112.9 | 95.6 | 103.3 | 56.7 | 115.2 | 100.9 | **117.7 ± 1.0** |
| **Average** | 54.6 | 61.6 | 60.6 | 70.4 | 70.3 | 64.4 | 66.8 | 67.7 | 80.0 | 76.5 | **83.6** |

**Predictive Capability.** In Subsection 4.3, we evaluate the predictive capability of the model trained with VIPO compared to a model trained using the original model loss, which does not account for value function inconsistency—the latter has been adopted by MOPO, MOReL, and MOBILE. The results on the D4RL benchmark indicate that VIPO achieves lower predictive error.

## 4.1 BENCHMARK RESULTS

In this subsection, we evaluate the performance of VIPO on the D4RL and NeoRL benchmarks. Specifically, we demonstrate its effectiveness by replacing the model in previous methods with the one learned by VIPO while keeping the policy extraction component (i.e., the planner) unchanged. For example, in the case of MOPO, we substitute its model with the one learned by VIPO and use the same planner, resulting in the VIPO-MOPO algorithm listed in Table 1. Compared to the retrained MOPO* results reported by (Sun et al., 2023) on the "v2" datasets (original results in (Yu et al., 2020) are trained on "v0" datasets and achieve an average score of 36.7), VIPO-MOPO achieves superior performance across all tasks.

For the VIPO algorithm results in Table 1, we employ Algorithm 1 with a planner same as MOBILE, which uses the soft actor-critic (SAC) (Haarnoja et al., 2018) framework to recover $\pi_{\text{out}}$. Details of the planner are provided in Algorithm 2 in the Appendix. Our findings indicate that VIPO outperforms previous results across all tasks, establishing a new SOTA. Therefore, the notable improvements observed with VIPO-MOPO over MOPO and VIPO over MOBILE provide empirical evidence that VIPO learns a more accurate model.

Additionally, VIPO's performance on the NeoRL benchmark (Table 2) achieves SOTA on most tasks. These results underscore VIPO's ability to learn a highly accurate model compared to prior methods.

To evaluate the performance of VIPO, we utilize several standard D4RL offline reinforcement learning benchmark tasks (Fu et al., 2020) from OpenAI Gym (Brockman, 2016) and the near-real-world NeoRL benchmark (Qin et al., 2022), both simulated using MuJoCo (Todorov et al., 2012). We defer the introduction of D4RL, NeoRL, and the compared baseline methods to Section E.1.

### 4.1.1 D4RL EXPERIMENT RESULTS

The D4RL experimental results are shown in Table 1. We report the normalized score for each task and the average performance over all tasks. These results are obtained during the final online evaluation conducted after training. Our results indicate that VIPO consistently outperforms all baseline methods across all tasks, achieving the highest average score compared to the other approaches. The MOPO results shown in Table 1 are labeled as MOPO*, as reported by (Sun et al., 2023), based on the "v2" datasets, whereas the original paper reported results based on "v0" datasets. For additional information, please refer to Appendix C.3 in (Sun et al., 2023).

The VIPO-MOPO implementation utilizes the same planner as MOPO, replacing their model with the one learned by VIPO. Similarly, the VIPO implementation adopts the same planner as MOBILE, substituting their model with ours. Therefore, the notable improvements observed with VIPO-MOPO

Table 2: Normalized average returns on NeoRL tasks. The experiments are run on MuJoCo-"v3" datasets over 4 random seeds. L = low, M = medium, H = High. We **bold** the highest mean.

| Task Name | BC | CQL | TD3+BC | EDAC | MOPO | MOBILE | VIPO(Ours) |
|-----------|-----|-----|--------|------|------|--------|------------|
| *halfcheetah-L* | 29.1 | 38.2 | 30.0 | 31.3 | 40.1 | 54.7 | **58.5 ± 0.1** |
| *hopper-L* | 15.1 | 16.0 | 15.8 | 18.3 | 6.2 | 17.4 | **30.7 ± 0.3** |
| *walker2d-L* | 28.5 | 44.7 | 43.0 | 40.2 | 11.6 | 37.6 | **67.6 ± 0.7** |
| *halfcheetah-M* | 49.0 | 54.6 | 52.3 | 54.9 | 62.3 | 77.8 | **80.9 ± 0.2** |
| *hopper-M* | 51.3 | 64.5 | **70.3** | 44.9 | 1.0 | 51.1 | 66.3 ± 0.2 |
| *walker2d-M* | 48.7 | 57.3 | 58.5 | 57.6 | 39.9 | 62.2 | **76.8 ± 0.1** |
| *halfcheetah-H* | 71.3 | 77.4 | 75.3 | 81.4 | 65.9 | 83.0 | **89.4 ± 0.6** |
| *hopper-H* | 43.1 | 76.6 | 75.3 | 52.5 | 11.5 | 87.8 | **107.7 ± 0.5** |
| *walker2d-H* | 72.6 | 75.3 | 69.6 | 75.5 | 18.0 | 74.9 | **81.7 ± 1.0** |
| **Average** | 45.4 | 56.1 | 54.5 | 50.7 | 28.5 | 60.7 | **73.3** |

over MOPO and VIPO over MOBILE provide empirical evidence that the Value Inconsistency approach results in more effective models compared to those trained with the original model loss function. For a more detailed comparative analysis of the VIPO's model learning performance, please refer to Section 4.3.

### 4.1.2 NEORL EXPERIMENT RESULTS

In our evaluation, we selected six offline RL algorithms as baseline methods: BC, CQL, TD3+BC, EDAC, MOPO, and MOBILE (refer to Section E.1). The experimental results are summarized in Table 2, where (L, M, H) denote three dataset types—low, medium, and high quality, respectively. Our proposed VIPO method demonstrates superior performance on most tasks. Notably, compared to the previous state-of-the-art method MOBILE, VIPO demonstrates significant performance improvements on specific tasks, including Walker2d-L, Hopper-L, and Hopper-H. This outstanding performance on the challenging NeoRL benchmark highlights the potential of our algorithm for real-world applications.

### 4.2 REVISITING UNCERTAINTY

In this experiment, we demonstrate that the model uncertainty learned by MOPO is unreliable due to the inaccuracies in its model. In contrast, VIPO can successfully capture the expected level of uncertainties.

### 4.2.1 UNCERTAINTY DEFINITION

In MOPO, the max-aleatoric uncertainty quantifier is utilized, with the model dynamics represented as $\hat{P}_k(s'|s,a) = \mathcal{N}(\mu_k(s,a), \Sigma_k(s,a))$, where $\mu_k$ and $\Sigma_k$ denote the mean and covariance matrix of the multivariate Gaussian modeling the $k$-th transition dynamics in the ensemble. The uncertainty is defined as

$$U(s,a) := \max_k \|\Sigma_k(s,a)\|_{\mathrm{F}}, \tag{12}$$

where $\|\cdot\|_{\mathrm{F}}$ denotes the Frobenius norm. As justified in MOPO, this uncertainty estimator captures both epistemic and aleatoric uncertainty of the true dynamics. To ensure a fair and consistent comparison, we adopt the same estimator to quantify the uncertainty of the models learned by both MOPO and VIPO.

### 4.2.2 EXPERIMENT SETUP

The experiment is conducted on the D4RL Walker2d-medium-replay dataset, which contains 301,698 transitions $(s, a, r, s')$. From this dataset, 1,000 transitions are randomly selected. For each selected $(s, a)$ pair, the dataset is searched to identify all other $(s', a')$ pairs that fall within the range $(s \pm 0.8, a \pm 0.8)$. These additional points, together with the original 1,000 points, form a candidate dataset, resulting in a total of 49,952 points. This approach allows us to capture data points with similar (s, a) values, which likely share similar contexts in the MDP.

To evaluate the robustness of uncertainty measures, dropout ratios ranging from 0% to 100% are applied to the candidate dataset in 10% increments. For each dropout ratio, a subset of points from

the candidate dataset is randomly removed, and the remaining points are merged back into the original dataset to create modified datasets. An ensemble of transition models is trained on each modified dataset by MOPO and VIPO, respectively, and the uncertainty in the regions corresponding to the perturbed $(s, a)$ pairs is evaluated separately. The uncertainty is calculated using Eq. (12) and averaged across the initially selected 1000 state-action pairs.

### 4.2.3 EXPERIMENT RESULTS

Our experiment is based on the premise that decreasing the amount of data should lead to higher uncertainty in a well-learned model, because limited information about the environment naturally entails greater uncertainty. To test this, we progressively drop portions of the candidate dataset and train models with MOPO and VIPO under identical settings. We use Eq. (12) to evaluate the uncertainties. As illustrated in Figure 2, VIPO's uncertainty grows consistently with the drop ratio, aligning with the expected trend, whereas MOPO's uncertainty remains largely insensitive to data reduction. This result highlights VIPO's potential to capture uncertainty more faithfully and to learn a more accurate model.

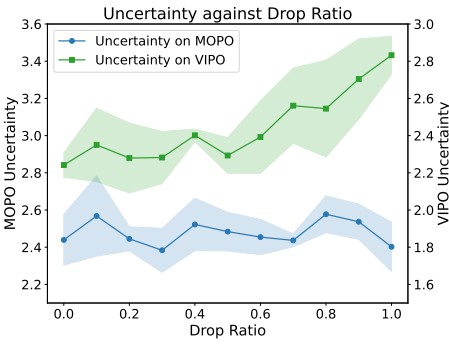

Figure 2: Uncertainty of models trained by MOPO and VIPO, averaged over 4 random seeds.

### 4.3 PREDICTIVE CAPABILITY

In this section, we train a model using only the original loss (referred to as OL Model in Table 3) defined in Eq. (7), which aligns with the approach adopted by previous methods such as MOPO, MOReL, and MOBILE. Subsequently, we train a model with VIPO. Finally, we assess the predictive accuracy of the OL Model and VIPO Model by comparing the predicted reward and next state $(\hat{r}, \hat{s}')$ with the true values $(r, s')$ for a given state-action pair $(s, a)$.

For all tasks, the dataset is split into training and validation sets in a 9:1 ratio. The model predictive error is measured as the mean squared error between the predicted and true values on the validation set. To ensure consistency, the same validation set is used for evaluating model performance across all tasks.

As summarized in Table 3, our results demonstrate that the VIPO achieves lower predictive error than the OL Model on the majority of tasks.

Table 3: Model predictive error comparison on D4RL Gym tasks, averaged over 6 random seeds.

| Task Name | OL Model Error | VIPO Model Error |
|---|---|---|
| *halfcheetah-r* | $0.079 \pm 0.008$ | $0.073 \pm 0.002$ |
| *hopper-r* | $0.0003 \pm 1\text{e-}4$ | $0.0003 \pm 1\text{e-}4$ |
| *walker2d-r* | $0.295 \pm 0.024$ | $0.247 \pm 0.013$ |
| *halfcheetah-m* | $0.550 \pm 0.060$ | $0.193 \pm 0.035$ |
| *hopper-m* | $0.009 \pm 0.002$ | $0.007 \pm 3\text{e-}4$ |
| *walker2d-m* | $0.287 \pm 0.122$ | $0.270 \pm 0.085$ |
| *halfcheetah-m-r* | $0.396 \pm 0.045$ | $0.289 \pm 0.023$ |
| *hopper-m-r* | $0.017 \pm 0.006$ | $0.014 \pm 0.002$ |
| *walker2d-m-r* | $0.285 \pm 0.020$ | $0.219 \pm 0.005$ |
| *halfcheetah-m-e* | $0.081 \pm 0.024$ | $0.070 \pm 0.010$ |
| *hopper-m-e* | $0.002 \pm 6\text{e-}4$ | $0.002 \pm 4\text{e-}4$ |
| *walker2d-m-e* | $0.077 \pm 0.002$ | $0.070 \pm 0.0026$ |
| **Average** | $0.173 \pm 0.026$ | $\mathbf{0.121 \pm 0.015}$ |

On average, VIPO reduces the model error from $0.173 \pm 0.026$ (OL Model) to $0.121 \pm 0.015$, reflecting a relative reduction of approximately 30%. This validates the effectiveness of incorporating the value function inconsistency loss into model training. These findings further confirm that VIPO can effectively learn a highly accurate model.

## 5 CONCLUSION

This paper proposes VIPO, a model-based offline RL algorithm that incorporates value function inconsistency into model training. Through extensive evaluations from multiple perspectives, we show that VIPO learns accurate models efficiently and consistently outperforms existing methods. As a general and readily applicable framework, VIPO can be integrated into model-based offline RL algorithms to improve model accuracy. Our work may encourage researchers to exploit data more comprehensively, rather than relying solely on costly architectures such as diffusion or transformers.

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

# Supplementary Material

## Table of Contents

## A  RELATED WORK

Offline RL focuses on learning effective policies solely from a pre-collected behavior dataset and has demonstrated significant success in practical applications (Rafailov et al., 2021; Singh et al., 2020; Li et al., 2010; Yu et al., 2018). Similar to online RL, offline RL has been explored using both model-free and model-based algorithms, distinguished by whether or not they involve learning a dynamics model.

**Model-free offline RL.** Existing model-free methods on offline RL can be roughly categorized into the following two taxonomies: pessimistic value-based methods and regularized policy-based methods. Pessimistic value-based approaches achieve conservatism by incorporating penalty terms into the value optimization objective, discouraging the value function from being overly optimistic on out-of-distribution (OOD) actions. Specifically, CQL (Kumar et al., 2020) applies equal penalization to Q-values for all OOD samples, whereas EDAC (An et al., 2021) and PBRL (Bai et al., 2022) adjust the penalization based on the uncertainty level of the Q-value, measured using a neural network ensemble. In comparison, regularized policy-based approaches constrain the learned policy to stay close to the behavior policy, thereby avoiding OOD actions. For instance, BEAR (Kumar et al., 2019) constrains the optimized policy by minimizing the MMD distance to the behavior policy. BCQ (Fujimoto et al., 2019) restricts the action space to those present in the dataset by utilizing a learned Conditional-VAE (CVAE) behavior-cloning model. Alternatively, TD3+BC (Fujimoto & Gu, 2021) simply adds a behavioral cloning regularization term to the policy optimization objective and achieves excellent performance across various tasks. IQL (Kostrikov et al., 2021) adopts an advantage-weighted behavior cloning approach, learning Q-value functions directly from the dataset. Meanwhile, DQL (Wang et al., 2022) leverages diffusion policies as an expressive policy class to enhance behavior-cloning.

**Model-based offline RL.** We focus on Dyna-style model-based RL (Janner et al., 2019), which learns a dynamics model from the dataset and uses it to augment the dataset with synthetic samples. However, due to inevitable model errors, conservatism remains crucial to prevent the policy from overgeneralizing to regions where the dynamics model predictions are unreliable. For example,

COMBO (Yu et al., 2021) extends CQL to a model-based setting by enforcing small Q-values for OOD samples generated by the dynamics model. RAMBO (Rigter et al., 2022) incorporates conservatism by adversarially training the dynamics model to minimize the value function while maintaining accurate transition predictions. Most model-based methods achieve conservatism through uncertainty quantification, penalizing rewards in regions with high uncertainty. Specifically, MOPO (Yu et al., 2020) uses the max-aleatoric uncertainty quantifier, MOReL (Kidambi et al., 2020) employs the max-pairwise-diff uncertainty quantifier, and MOBILE (Sun et al., 2023) leverages the Model-Bellman inconsistency uncertainty quantifier. In this work, we achieve conservatism by incorporating the value function inconsistency loss, enabling the training of a more reliable model.

## B    PROOF OF THE MODEL GRADIENT THEOREM

*Proof.* The gradient of the original loss can be obtained through automatic differentiation. Now we focus on the gradient derivation of the value consistency loss as defined in Eq. (6).

$$
\nabla_\theta \mathbb{E}_{s \sim \rho_0} \left[ \left( V_d^\mu(s) - V_m^\mu(s) \right)^2 \right] = -2 \mathbb{E}_{s \sim \rho_0} \left[ \left( V_d^\mu(s) - V_m^\mu(s) \right) \nabla_\theta V_m^\mu(s) \right]. \tag{13}
$$

Then, we need to calculate the gradient of $V_m^\mu(s)$ in terms of $\theta$. To achieve this goal, we start by decomposing the value function using the Bellman equation, which is inspired by the previous work (Rigter et al., 2022). For the given state $s$:

$$
\begin{aligned}
V_m^\mu(s) &= \sum_a \mu(a \mid s) Q_m(s, a) \\
&= \sum_a \mu(a \mid s) \sum_{s', r \sim P_\theta} \left( r + \gamma V_m^\mu(s') \right) \cdot P_\theta(s', r \mid s, a),
\end{aligned} \tag{14}
$$

where $Q_m(s, a)$ is the state-action value function under dynamics model $P_\theta$ and the behavior policy $\mu$. Applying the product rule:

$$
\begin{aligned}
\nabla_\theta V_m^\mu(s) &= \sum_a \mu(a \mid s) \sum_{s', r} \left[ \left( r + \gamma V_m^\mu(s') \right) \cdot \nabla_\theta P_\theta(s', r \mid s, a) \right. \\
&\quad \left. + P_\theta(s', r \mid s, a) \cdot \nabla_\theta \left( r + \gamma V_m^\mu(s') \right) \right] \\
&= \sum_a \mu(a \mid s) \sum_{s', r} \left( r + \gamma V_m^\mu(s') \right) \cdot \nabla_\theta P_\theta(s', r \mid s, a) \\
&\quad + \gamma \sum_a \mu(a \mid s) \sum_{s'} P_\theta(s' \mid s, a) \cdot \nabla_\theta V_m^\mu(s').
\end{aligned} \tag{15}
$$

Define $\psi(s) = \sum_a \mu(a|s) \sum_{s', r} (r + \gamma V_m^\mu(s')) \cdot \nabla_\theta P_\theta(s', r|s, a)$, and additionally define $\rho_\theta^\mu(s \to x, n)$ as the transition probability under the policy $\mu$ from state $s$ to $x$ after $n$ steps in the learned MDP model $P_\theta$. Then, we can rewrite the above equation as:

$$
\begin{aligned}
\nabla_\theta V_m^\mu(s) &= \psi(s) + \gamma \sum_{s'} \rho_\theta^\mu(s \to s', 1) \nabla_\theta V_m^\mu(s') \\
&= \psi(s) + \gamma \sum_{s'} \rho_\theta^\mu(s \to s', 1) \left[ \psi(s') + \gamma \sum_{s''} \rho_\theta^\mu(s' \to s'', 1) \nabla_\theta V_m^\mu(s'') \right] \\
&= \psi(s) + \gamma \sum_{s'} \rho_\theta^\mu(s \to s', 1) \psi(s') + \gamma^2 \sum_{s''} \rho_\theta^\mu(s \to s'', 2) \nabla_\theta V_m^\mu(s'') \\
&= \sum_{s'} \sum_{t=0}^{\infty} \gamma^t \rho_\theta^\mu(s \to s', t) \psi(s'),
\end{aligned} \tag{16}
$$

where the last equation of Eq. (16) is obtained by continuing to unroll $\nabla_\theta V(\cdot)$.

Substituting Eq. (16) into Eq. (13), we have

$$
\begin{aligned}
\nabla_\theta \mathbb{E}_{s\sim\rho_0} &\left[ \left( V_d^\mu(s) - V_m^\mu(s) \right)^2 \right] \\
&= -2\mathbb{E}_{s\sim\rho_0} \left[ \left( V_d^\mu(s) - V_m^\mu(s) \right) \nabla_\theta V_m^\mu(s) \right] \\
&= -2\mathbb{E}_{s\sim\rho_0} \left[ \left( V_d^\mu(s) - V_m^\mu(s) \right) \sum_{s'} \sum_{t=0}^\infty \gamma^t \rho_\theta^\mu(s \to s', t) \psi(s') \right] \\
&= -2\sum_{s\in\mathcal{S}} \rho_0(s) \left( V_d^\mu(s) - V_m^\mu(s) \right) \sum_{s'} \sum_{t=0}^\infty \gamma^t \rho_\theta^\mu(s \to s', t) \psi(s') \\
&= -2\sum_{s\in\mathcal{S}} \sum_{s'\in\mathcal{S}} \rho_0(s) \left( V_d^\mu(s) - V_m^\mu(s) \right) d_\theta^\mu(s, s') \psi(s'),
\end{aligned}
\tag{17}
$$

where $d_\theta^\mu$ is the (improper) discounted state transition probability from $s$ to $s'$. Note that from the definition of $\psi(s)$, we have

$$
\begin{aligned}
\psi(s) &= \sum_a \mu(a|s) \sum_{s',r} (r + \gamma V_m^\mu(s')) \cdot \nabla_\theta P_\theta(s', r|s, a) \\
&= \sum_a \mu(a|s) \sum_{s',r} (r + \gamma V_m^\mu(s')) P_\theta(s', r|s, a) \frac{\nabla_\theta P_\theta(s', r|s, a)}{P_\theta(s', r|s, a)} \\
&= \mathbb{E}_{a\sim\mu, (r,s')\sim P_\theta} \left[ (r + \gamma V_m^\mu(s')) \nabla_\theta \log P_\theta(s', r|s, a) \right].
\end{aligned}
$$

Therefore, we have

$$
\psi(s') = \mathbb{E}_{a'\sim\mu, (r',s'')\sim P_\theta} \left[ (r' + \gamma V_m^\mu(s'')) \nabla_\theta \log P_\theta(s'', r'|s', a') \right]
$$

Plugging $\psi(s')$ into Eq. (17), we obtain

$$
\begin{aligned}
\nabla_\theta \mathbb{E}_{s\sim\rho_0} \left[ \left( V_d^\mu(s) - V_m^\mu(s) \right)^2 \right] &= -2\sum_{s\in\mathcal{S}} \sum_{s'\in\mathcal{S}} \rho_0(s) \left( V_d^\mu(s) - V_m^\mu(s) \right) d_\theta^\mu(s, s') \psi(s') \\
&= -2\mathbb{E}_{s\sim\rho_0, s'\sim d_\theta^\mu, a'\sim\mu, (r',s'')\sim P_\theta} \left[ \left( V_d^\mu(s) - V_m^\mu(s) \right) \cdot \right. \\
&\qquad \left. (r' + \gamma V_m^\mu(s'')) \nabla_\theta \log P_\theta(s'', r'|s', a') \right].
\end{aligned}
$$

Hence, we finish the proof. $\qquad\square$

## C    PROOF OF PROPOSITIONS

**Proof of Theorem 3.1**

*Proof.* We want to show there is exactly one function $V^*$ satisfying

$$
V^*(s) = \widehat{T}_d^\mu V^*(s)
$$

for any state $s$, which means $V^*$ is the unique fixed point of $\widehat{T}_d^\mu$.

In the following, We prove that $\widehat{T}_d^\mu$ is a $\gamma$-contraction with respect to the supremum norm. Let $\|V\|_\infty = \sup_s |V(s)|$ denote the sup norm of a value function $V$. We will show

$$
\|\widehat{T}_d^\mu V - \widehat{T}_d^\mu W\|_\infty \le \gamma \|V - W\|_\infty \quad \text{for all } V, W.
$$

Fix any state $s$, we consider the difference $\widehat{T}_d^\mu V(s) - \widehat{T}_d^\mu W(s)$.

If $|\mathcal{D}_s| > 0$, then

$$\widehat{T}_d^\mu V(s) \;=\; \frac{1}{|\mathcal{D}_s|} \sum_{(s,a,r,s')\in\mathcal{D}_s} \big[r + \gamma\,V(s')\big],$$

$$\widehat{T}_d^\mu W(s) \;=\; \frac{1}{|\mathcal{D}_s|} \sum_{(s,a,r,s')\in\mathcal{D}_s} \big[r + \gamma\,W(s')\big].$$

It follows that

$$\widehat{T}_d^\mu V(s) - \widehat{T}_d^\mu W(s) \;=\; \frac{\gamma}{|\mathcal{D}_s|} \sum_{(s,a,r,s')\in\mathcal{D}_s} \big[V(s') - W(s')\big].$$

Therefore, we have

$$\big|\widehat{T}_d^\mu V(s) - \widehat{T}_d^\mu W(s)\big| \;=\; \frac{\gamma}{|\mathcal{D}_s|} \Big|\sum_{(s,a,r,s')\in\mathcal{D}_s} \big[V(s') - W(s')\big]\Big|$$

$$\leq\; \frac{\gamma}{|\mathcal{D}_s|} \sum_{(s,a,r,s')\in\mathcal{D}_s} \big|V(s') - W(s')\big|.$$

Since $\big|V(s') - W(s')\big| \leq \|V - W\|_\infty$ for all $s'$, it holds that

$$\big|\widehat{T}_d^\mu V(s) - \widehat{T}_d^\mu W(s)\big| \;\leq\; \frac{\gamma}{|\mathcal{D}_s|} \sum_{(s,a,r,s')\in\mathcal{D}_s} \|V - W\|_\infty \;=\; \gamma\,\|V - W\|_\infty.$$

If $|\mathcal{D}_s| = 0$, by definition, we have

$$\big|\widehat{T}_d^\mu V(s) - \widehat{T}_d^\mu W(s)\big| \;=\; 0 \;\leq\; \gamma\,\|V - W\|_\infty.$$

Overall, for any state $s$, we have

$$\big|\widehat{T}_d^\mu V(s) - \widehat{T}_d^\mu W(s)\big| \;\leq\; \gamma\,\|V - W\|_\infty.$$

Taking the supremum over $s$ yields

$$\|\widehat{T}_d^\mu V - \widehat{T}_d^\mu W\|_\infty \;=\; \sup_s \big|\widehat{T}_d^\mu V(s) - \widehat{T}_d^\mu W(s)\big| \;\leq\; \gamma\,\|V - W\|_\infty.$$

Hence $\widehat{T}_d^\mu$ is indeed a $\gamma$-contraction under the sup norm.

By the Banach Fixed Point Theorem, any $\gamma$-contraction ($0 \leq \gamma < 1$) on a complete normed vector space has a unique fixed point. Therefore, there exists a unique $V^*$ such that

$$V^*(s) \;=\; \widehat{T}_d^\mu V^*(s), \quad \forall s.$$

We denote $V^*(s)$ by $V_d^\mu(s)$ which completes our proof. $\qquad\square$

**Proof of Theorem 3.2**

*Proof.* This follows the same argument as the standard proof of the Bellman operator's fixed-point uniqueness, substituting the true model $P$ with the learned model $P_\theta$. $\qquad\square$

## D   PLANNER DETAILS

To implement Algorithm 1, we adopt the objective function defined in (Janner et al., 2019; Chua et al., 2018) as the original model loss. The specific form of the original model loss is expressed as:

$$\mathcal{L}_{ori}(\theta) = -\mathbb{E}_{\mathcal{D}}\left[\log P_\theta(s', r | s, a)\right] \tag{18}$$

For instance, we might define our model predictive model $P_\theta$ to produce a Gaussian distribution with diagonal covariances, parameterized by $\theta$ and conditioned on $s$ and $a$, i.e.: $P_\theta(s', r | s, a) = \mathcal{N}(\boldsymbol{\mu}_\theta(s, a), \boldsymbol{\Sigma}(s, a))$. Then the original model loss has the following form:

$$\mathcal{L}_{ori}^G(\theta) = \sum_{n=1}^{N} [\boldsymbol{\mu}_\theta(s_n, a_n) - s_{n+1}] \boldsymbol{\Sigma}_\theta^{-1}(s_n, a_n)[\boldsymbol{\mu}_\theta(s_n, a_n) - s_{n+1}] + \log\det\boldsymbol{\Sigma}_\theta(s_n, a_n). \tag{19}$$

---

**Algorithm 2** Planner

---

1: **Require:** Offline dataset $\mathcal{D} = \{(s_i, a_i, r_i, s'_i)\}_{i=1}^N$; approximate dynamics model $P_\theta$ learned from Algorithm 1; critics $\{Q_{\psi_1}, Q_{\psi_2}\}$
2: Initilize the replay buffer $\mathcal{D}_m \leftarrow \emptyset$
3: **while** not reaching maximum iterations $T$ or convergence criterion **do**
4:     Generate $h$-step rollouts by $P_\theta$ and add them to $\mathcal{D}_m$
5:     Sample a mini-batch $B = \{s, a, r, s'\}$ from $\mathcal{D} \cup \mathcal{D}_m$
6:     Compute the target values for $B$ according to Eq. (22) and Eq. (23)
7:     Learn the optimal control policy $\pi_\phi$ according to Eq. (24)
8: **end while**
9: Output the optimized policy parameter $\phi^* = \phi$

---

Table 4: Hyperparameters of Policy Optimization in VIPO.

| Hyperparameters | Value | Description |
|---|---|---|
| K | 2 | The number of critics. |
| Policy network | FC(256, 256) | Fully Connected (FC) layers with ReLU activations. |
| Q-network | FC(256, 256) | Fully Connected (FC) layers with ReLU activations. |
| actor learning rate | $1e-4$ | Policy learning rate. |
| critic learning rate | $3e-4$ | Critic learning rate. |
| value learning rate | $1e-4$ | Value network learning rate. |
| $\tau$ | $5e-3$ | Target network smoothing coefficient. |
| $\gamma$ | 0.99 | Discount factor. |
| Optimizer | Adam | Optimizers of the actor, critic, and value networks. |
| Batch size | 256 | Batch size for each gradient update. |
| $N_{\text{iter}}$ | 3M | Total gradient steps. |

We use Algorithm 2 as a concrete implementation of the PLANNER in Algorithm 1. Algorithm 2 is adapted from Algorithm 1 in (Sun et al., 2023). Given a pre-trained environment model $P_\theta$ generated by Algorithm 1, the agent simulates $h$-step rollouts starting from the state in $\mathcal{D}$ in the learned model $P_\theta$ and then stores these synthetic transitions to the replay buffer $\mathcal{D}_m$. For policy training, we incorporate the uncertainty quantification $\mathcal{U}(s, a)$ with the soft actor-critic (SAC) algorithm (Haarnoja et al., 2018). Specifically, we use the uncertainty quantification based on Bellman Inconsistency proposed in (Sun et al., 2023):

$$
\begin{aligned}
\mathcal{U}(s, a) &= \text{Std}\left(P_\theta^i Q_\psi(s, a)\right) \\
&= \text{Std}\left(\gamma \mathbb{E}_{\substack{\{s'_j\} \sim P_\theta^i \\ \{a'_j\} \sim \pi}}\left[\min_{k=1,2} \bar{Q}_{\psi_k}(s'_j, a'_j)\right]\right),
\end{aligned}
\tag{20}
$$

where $\bar{Q}_{\psi_k}$ is the target state-action value network obtained by an exponentially moving average of parameters of the state-action value network $Q_{\psi_k}$. The objective function for critics is defined as:

$$
\mathcal{L}_{critic} = \mathbb{E}_{(s,a,r,s') \sim \mathcal{D} \cup \mathcal{D}_m}\left[(Q_{\psi_k} - y)^2\right],
\tag{21}
$$

where the target value for $(s, a, r, s') \in \mathcal{D}$ is

$$
y = r + \gamma\left[\min_{k=1,2} \bar{Q}_{\psi_k}(s', a') - \alpha \log \pi_\phi(a'|s')\right],
\tag{22}
$$

and the target for $(s, a, r, s') \in \mathcal{D}_m$ is

$$
y = r + \gamma\left[\min_{k=1,2} \bar{Q}_{\psi_k}(s', a') - \alpha \log \pi_\phi(a'|s')\right] - \beta \mathcal{U}(s, a).
\tag{23}
$$

The policy is optimized by solving the following optimization problem:

$$
\pi_\phi = \max_\phi \mathbb{E}_{\substack{\sim \mathcal{D} \cup \mathcal{D}_m \\ a \sim \pi_\phi}}\left[\min_{k=1,2} Q_{\psi_k}(s, a) - \alpha \log \pi_\phi(a|s)\right]
\tag{24}
$$

Table 5: Fine-tuned hyperparameters of VIPO.

| Domain Name | Task Name | $\lambda$ | $h$ | $\eta$ | $\beta$ |
|---|---|---|---|---|---|
| Gym | halfcheetah-random | 0.35 | 5 | 0.05 | 0.5 |
| | hopper-random | 0.55 | 5 | 0.05 | 5.0 |
| | walker2d-random | 0.3 | 5 | 0.5 | 0.5 |
| | halfcheetah-medium | 0.35 | 5 | 0.05 | 0.5 |
| | hopper-medium | 0.45 | 5 | 0.05 | 1.5 |
| | walker2d-medium | 0.3 | 2 | 0.5 | 0.5 |
| | halfcheetah-medium-replay | 0.35 | 5 | 0.05 | 0.1 |
| | hopper-medium-replay | 0.8 | 5 | 0.05 | 0.1 |
| | walker2d-medium-replay | 0.3 | 1 | 0.5 | 0.5 |
| | halfcheetah-medium-expert | 0.35 | 5 | 0.05 | 1.0 |
| | hopper-medium-expert | 0.55 | 5 | 0.05 | 5.0 |
| | walker2d-medium-expert | 0.3 | 1 | 0.5 | 2.0 |
| NeoRL | HalfCheetah-L | 0.4 | 5 | 0.05 | 0.5 |
| | Hopper-L | 0.03 | 5 | 0.5 | 2.5 |
| | Walker2d-L | 0.35 | 1 | 0.5 | 2.5 |
| | HalfCheetah-M | 0.4 | 5 | 0.05 | 0.5 |
| | Hopper-M | 0.01 | 5 | 0.5 | 1.5 |
| | Walker2d-M | 0.35 | 1 | 0.5 | 2.5 |
| | HalfCheetah-H | 0.4 | 5 | 0.05 | 1.5 |
| | Hopper-H | 0.03 | 5 | 0.5 | 2.5 |
| | Walker2d-H | 0.35 | 1 | 0.5 | 2.5 |

# E  EXPERIMENTAL DETAILS

## E.1  EXPERIMENTAL SETUPS

**D4RL.** The D4RL Mujoco tasks serve as a benchmark for evaluating offline RL algorithms in continuous control environments, such as Hopper, Walker2d, and HalfCheetah. These tasks employ datasets of varying quality, including random, expert, and mixed trajectories, to evaluate agents' ability to derive effective policies from offline data. This setup provides a standardized framework for assessing offline RL performance in continuous control domains.

**NeoRL.** NeoRL's MuJoCo tasks provide offline RL benchmarks in environments such as HalfCheetah-v3, Walker2d-v3, and Hopper-v3. These tasks are based on conservative datasets generated from suboptimal policies, mimicking real-world scenarios characterized by limited and narrowly distributed data. This setup poses a challenge for algorithms to learn effective policies from constrained offline datasets, promoting the development of methods applicable to practical settings. Notably, NeoRL offers varying numbers of trajectories for training data across tasks: 100, 1,000, and 10,000. For our experiments, we uniformly selected 1,000 trajectories for each task.

**Baselines.** In D4RL tasks, we evaluate VIPO against various offline RL algorithms, including model-free methods such as TD3+BC (Fujimoto & Gu, 2021), which trains a policy subject to a behavior cloning constraint. We also compare VIPO to CQL (Kumar et al., 2020), which calculates conservative Q-values for out-of-distribution (OOD) samples, and IQL (Kostrikov et al., 2021), which learns optimal policies offline through expectile regression on the dataset to address extrapolation errors. Additionally, we consider DQL (Wang et al., 2022), which utilizes diffusion models to effectively generate optimal actions from offline datasets. In the model-based category, we include MOPO (Yu et al., 2020), which incorporates uncertainty-aware dynamics models to penalize OOD state-action pairs; MOReL (Kidambi et al., 2020), which constructs uncertainty-aware, penalized MDPs to restrict policy learning to in-distribution states; RAMBO (Rigter et al., 2022), which uses adversarial models to handle distributional shifts in offline data; and MOBILE (Sun et al., 2023), which applies Bellman-inconsistency uncertainty quantification for robust offline policy learning.

For the NeoRL tasks, we compare VIPO against BC, CQL, TD3+BC, EDAC, MOPO, and MOBILE. BC (Behavioral Cloning) learns by directly imitating the data-generating policy through supervised learning. EDAC (An et al., 2021) addresses value overestimation by decoupling the target policy from the behavior policy.

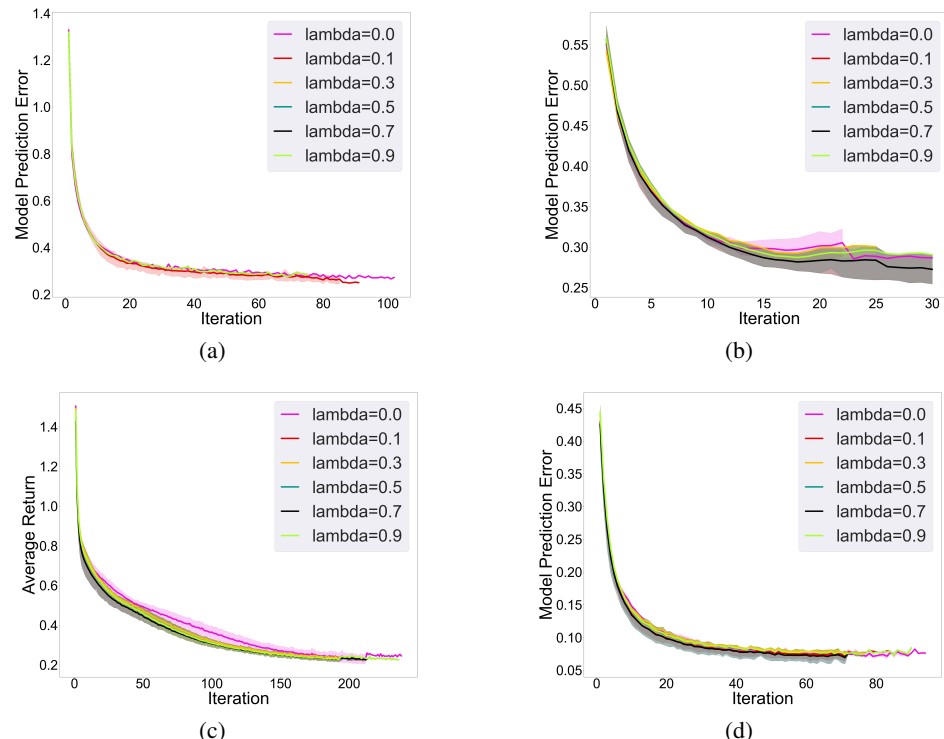

Figure 3: The model prediction error on four walker2d datasets of the D4RL task: (a) walker2d-random-v2; (b) walker2d-medium-v2; (c) walker2d-medium-replay-v2; (d) walker2d-medium-expert-v2.

## E.2 BENCHMARKS

We perform experiments on Gym tasks (v2 version) included in the D4RL (Fu et al., 2020) benchmark. In addition, we leverage the NeoRL benchmark, which offers a more challenging evaluation setting that closely resembles real-world scenarios, to provide a more comprehensive assessment of offline RL algorithms. NeoRL tasks are constructed using conservative datasets generated from suboptimal policies, reflecting real-world conditions characterized by limited and narrowly distributed data. This design poses substantial challenges for algorithms to learn effective policies from constrained offline datasets, thereby driving the development of approaches better suited for practical applications.

The following sections outline the sources of the reported performance on these benchmarks.

**D4RL.** (1) For MOPO* (Yu et al., 2020), as the original paper reported results on "v0" datasets, we reference the experimental results provided in (Sun et al., 2023), which are based on the "v2" datasets. For CQL (Kumar et al., 2020), we report the scores obtained on the "v2" datasets using the codebase available at `github.com/yihaosun1124/OfflineRL-Kit`. (2) For TD3+BC (Fujimoto & Gu, 2021), MOReL (Kidambi et al., 2020), COMBO (Yu et al., 2021), RAMBO (Rigter et al., 2022), and MOBILE (Sun et al., 2023), we directly cite the performance results reported in their respective original papers, as these studies evaluated Gym tasks using the "v2" datasets (refer to Table 1). (3) For IQL (Kostrikov et al., 2021) and DQL (Wang et al., 2022), we conducted experiments on the "v2" random datasets using the codebase provided by the authors of the respective papers. For other "v2" datasets, the results are taken directly from their original publications.

**NeoRL.** The performance results for BC, CQL, and MOPO are sourced from the original NeoRL paper. For TD3+BC and EDAC, we report the scores retrained by the authors of (Sun et al., 2023).

## E.3 HYPERPARAMETERS

The hyperparameters selected for each task are detailed in Table 4. We list the hyperparameters that we finetuned as follows. All experiments were conducted using NVIDIA RTX 4090 GPUs.

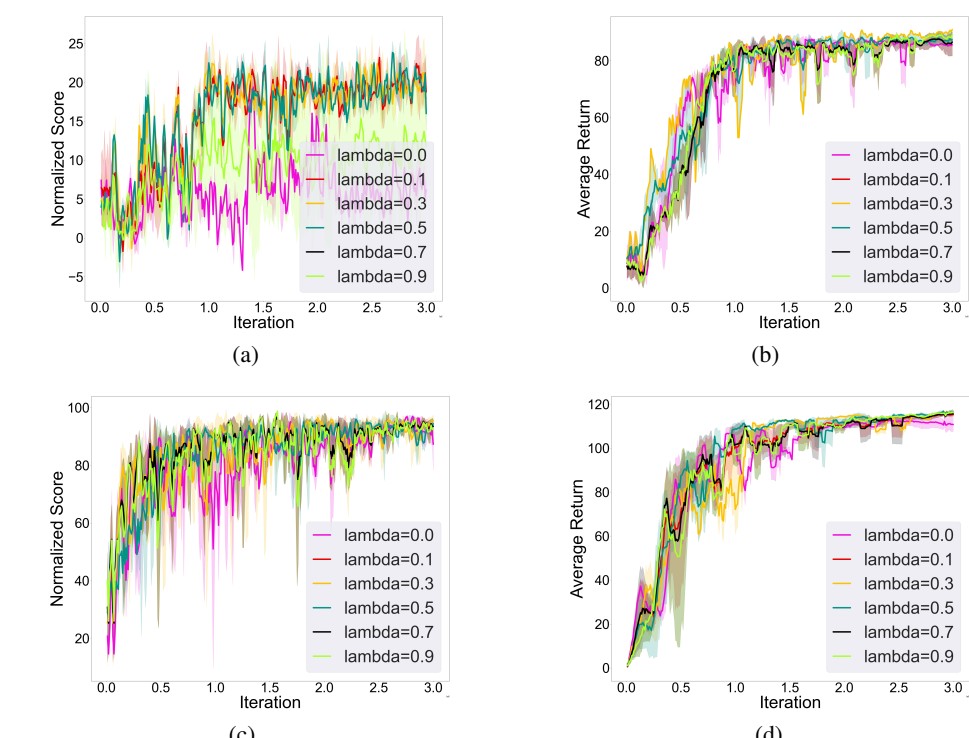

Figure 4: The policy training process on four walker2d datasets of the D4RLtask: (a) walker2d-random-v2; (b) walker2d-medium-v2; (c) walker2d-medium-replay-v2; (d) walker2d-medium-expert-v2.

**Balance Coefficient $\lambda$:** For Gym tasks, the balance coefficient $\lambda$ is optimized by searching within the range $\{0.1, 0.2, 0.3, 0.35, 0.4, 0.45, 0.5, 0.55, 0.6, 0.7, 0.8, 0.9\}$. For NeoRL tasks, the search range is restricted to $\{0.001, 0.01, 0.03, 0.1, 0.2, 0.3, 0.35, 0.4, 0.45, 0.5, 0.55, 0.6, 0.65\}$. Notably, in the hopper task, a smaller value of $\lambda$ has been found to result in better performance.

**Rollout Length $h$:** VIPO employs short-horizon rollouts, consistent with methods like MOPO and MOBILE. The rollout length $h$ is fine-tuned within the range $\{1, 2, 5, 7\}$ for both Gym and NeoRL tasks.

**Real Ratio $\eta$:** The real ratio $\eta$ defines the proportion of data sampled from the offline dataset in each batch. Specifically, $\eta$ is tuned within the range $\{0.05, 0.5\}$.

**Penalty Coefficient $\beta$:** For Gym tasks, $\beta$ is fine-tuned within the range $\{0.1, 0.5, 1.0, 1.5, 2.0, 2.5\}$. For NeoRL tasks, the tuning range for $\beta$ is $\{0.5, 1.0, 1.5, 2.0, 2.5\}$.

The fine-tuned hyperparameters for each task are summarized in Table 5.

# F    ABLATION STUDY

In this section, we conduct ablation experiments to analyze the effect of the hyperparameter $\lambda$ in Eq. (8) on four Walker2D datasets from the D4RL benchmark. Note that when training a model, the algorithm may stop at different stages, depending on the specific algorithm convergence judgment conditions. Specifically, in addition to limiting the maximum number of gradient updates, we also limit the number of invalid gradient updates, i.e., if the model prediction error is reduced by less than 1%, the gradient update is considered invalid. The experimental results, presented in Fig. 3, indicate that $\lambda$ significantly influences the model dynamics learning process. Selecting an appropriate value of $\lambda$ leads to a model with reduced prediction error. Figure 4 illustrates the policy training process using models trained with different values of $\lambda$. The results suggest that VIPO facilitates learning a policy that achieves a higher score.

# G  MORE EXPERIMENTS ON ADROIT TASKS

Table 6: Normalized average returns on D4RL Adroit tasks, averaged over 4 random seeds.

| Task Name | BC | CQL | TD3+BC | EDAC | MOPO | MOBILE | VIPO(Ours) |
|---|---|---|---|---|---|---|---|
| *pen-human* | 25.8 ± 8.8 | 35.2 ± 6.6 | -1.0 | 52.1 ± 8.6 | 10.7 | 30.1 ± 14.6 | **52.6 ± 7.7** |
| *door-human* | 2.8 ± 0.7 | 1.2 ± 1.8 | -0.2 | **10.7 ± 6.8** | -0.2 | -0.2 ± 0.1 | 2.0 ± 0.3 |
| *hammer-human* | **3.1 ± 3.2** | 0.6 ± 0.5 | 0.2 | 0.8 ± 0.4 | 0.3 | 0.4 ± 0.2 | 1.1 ± 0.9 |
| *pen-cloned* | 38.3 ± 11.9 | 27.2 ± 11.3 | -2.1 | 68.2 ± 7.3 | 54.6 | 69.0 ± 9.3 | **71.1 ± 9.5** |
| *hammer-cloned* | 0.7 ± 0.3 | 1.4 ± 2.1 | -0.1 | 0.3 ± 0.0 | 0.5 | 1.5 ± 0.4 | **2.1 ± 0.2** |
| **Average** | 14.1 ± 5.0 | 13.1 ± 4.5 | -0.6 | **26.4 ± 4.6** | 13.2 | 20.2 ± 4.9 | 25.8 ± **3.7** |

The Adroit benchmark presents a set of high-dimensional manipulation tasks using a 24-DoF simulated robotic hand. The tasks include writing with a pen, hammering a nail, and opening a door, each requiring fine-grained control and coordination. We consider two types of datasets: "human", which consists of 25 expert demonstrations collected from real human teleoperation, and "cloned", a 50-50 mixture of these demonstrations and data generated by a cloned policy trained on them.

In Table 6, we report the evaluation results of VIPO on the Adroit benchmark from D4RL. The inherent complexity of the Adroit benchmark arises from its high-dimensional observation and action spaces combined with relatively sparse demonstration data. VIPO addresses these challenges by incorporating a value-informed loss into the dynamics model training, encouraging consistency between the predicted values under the learned dynamics and the empirical returns from the dataset. This promotes more value-aligned dynamics without explicitly optimizing the policy.

The results illustrate that VIPO consistently outperforms several established methods, including model-free algorithms (BC, CQL, TD3+BC) and prominent model-based approaches (MOPO, MO-BILE) across multiple benchmark tasks (see Table 6). Specifically, notable performance gains are observed in tasks such as pen-human (52.6 ± 7.7), pen-cloned (71.1 ± 9.5), and hammer-cloned (2.1 ± 0.2), underscoring VIPO's capacity to manage the intricacies associated with high-dimensional manipulation scenarios.

# H  DECLARATION

I declare that Large Language Models (LLMs) were used solely for language polishing in this paper. No other usage of LLMs was involved.

