# OpenReview forum: "VIPO: Value Function Inconsistency Penalized Offline Reinforcement Learning"
_ICLR.cc/2026/Conference — Submitted to ICLR 2026_

### Official Review · Reviewer_jZLG · 2025-10-28

**Soundness:** 3
**Presentation:** 3
**Contribution:** 2
**Rating:** 4
**Confidence:** 4

**Summary:**

This paper presents a novel offline model-based reinforcement learning method called VIPO that incorporates value inconsistency loss in the offline model learning, where two distinct values are learned from the same offline dataset by introducing dual usage of the offline data. Authors contend that the uncertainty estimation of the conventional offline model-based methods is hard to rely on when complex offline data is given to the agent. They further suggest a two-way value learning approach: learning a value solely learned from the offline data and another value function learned with the empirical transition dynamics model, supported by theoretical analysis and empirical observation. They conduct an empirical study with model-free and model-based offline RL baselines on the D4RL and NeoRL benchmarks. VIPO achieves superior performance across benchmarks, highlighting the lower model prediction error and higher empirical performance.

**Strengths:**

- The authors provide comprehensive explanations on connecting theoretical foundations and practical implementations.
- The authors rigorously provide supplementary proofs for theorems on value learning and the model gradient theorem.
- Experiments on D4RL show improved performance across model-free and model-based baselines.
- Experiments on NeoRL demonstrate an interesting perspective of VIPO when the given data is limited.

**Weaknesses:**

- The novelty of VIPO is limited. The main contribution relies on reducing uncertainty estimations to learn an accurate model by introducing an auxiliary value inconsistency loss into the original model learning. However, there is no guarantee that the learned model predicts reliable synthetic rollouts under out-of-support data in the current methodology. Besides, the value inconsistency loss largely depends on the assumption that the learned values precisely estimate ($V^\mu_d (s)$ and $V^\mu_m (s)$) approximate the true value(L171, L188), whereas the learned value function fails to predict accurate values when the given $(s,a)$ pair lies outside of the offline data in general [1]. Based on my conjecture, employing a surrogate objective with heuristics in Section 3.3 contributes to the empirical performance gains to some extent, while those observations benefit the task-specific problem formulation- MuJoCo locomotion tasks often exhibit tightly bounded states and actions.
- Experimental results present conflicting views from the perspective of the authors. In Table 1, MOPO outperforms VIPO-MOPO in five out of twelve cases when comparing the effect of planners, which contrasts with the authors' argument that VIPO-MOPO outperforms MOPO across all tasks in L353. Additionally, the y-axis ticks of Figure 2 are misaligned across baselines. While it is notable that VIPO demonstrates increasing uncertainty when the drop ratio increases, the absolute difference is marginal when the y-axis is scaled with each other.

[1] Levine, Sergey, et al. "Offline reinforcement learning: Tutorial, review, and perspectives on open problems." arXiv 2020.

**Questions:**

- Could you conduct additional experiments in more complex domains to verify that the empirical gains do not solely stem from the practical designs? Current tasks often contain narrow bounds in state or action spaces, which aligns with heuristics that numerical differences are trivial. For instance, the tabletop robotic manipulation benchmark [1] provides tasks with a wide-ranging state space (e.g., distance in Cartesian space from a hand to an object).
- Are there alternative ways to approximate the augmented model learning loss instead of replicating the same tuple for the next time-step tuple, which can be further extended to domains outside of locomotion tasks? I believe suggesting more comprehensive directions for implementing the model gradient theorem would significantly improve the novelty of the paper.
- What is the limitation of VIPO? Discussions on the potential drawbacks of the proposed method should be addressed.

[1] Mandlekar, Ajay, et al. "What matters in learning from offline human demonstrations for robot manipulation." arXiv 2021.

Minor problems
- L122: Maybe a typo? $P(s'|s,a) : \mathcal{S} \times \mathcal{A} \rightarrow \mathcal{S}$
- L170: What is the meaning of "densely sampled"? Does this sentence stand for a nearly full-coverage offline dataset?
- Table 1: Bold faces do not denote the best scores. (MOReL is the best in *hopper-r* and *walker2d-r*, VIPO-MOPO is the best in *hopper-m-e*)

---

> ### Author Response · Authors · 2025-11-21
>
> (W1: On the novelty of VIPO)
>
> We thank the reviewer for the comments and address the concerns as follows.
>
> 1) On the novelty of VIPO. VIPO does not simply tweak uncertainty estimation; it introduces a different mechanism for improving model learning. Prior model-based offline RL methods rely on ensemble-based uncertainty estimates and then apply heuristic penalties to predicted next states. In contrast, VIPO leverages the dataset in two complementary ways (Sec. 3.1): (i) learning a value function via MSBE minimization, and (ii) learning a model-based value function via Bellman evaluation. Their discrepancy provides a self-supervised training signal that, to the best of our knowledge, has not been exploited in previous model-based offline RL methods. Theorem 3.3 further gives an analytical gradient for the augmented loss, which is qualitatively different from the heuristic penalty designs used in MOPO, MOReL, COMBO, and MOBILE.
>
> We also note that multiple reviewers independently characterized the “value inconsistency’’ idea as novel and distinct from prior work, which supports our claim that the mechanism introduced by VIPO is conceptually new rather than a minor variant of existing uncertainty penalties.
>
> 2) On predicting accurate values outside the offline data. We agree that no offline RL method can accurately evaluate value functions on truly out-of-distribution $(s,a)$ pairs, simply because no data exist in those regions. This limitation is fundamental and not specific to VIPO. In our implementation, VIPO does not require accurate value estimates on arbitrary OOD pairs: both $V_d$ (Eq. (9)) and $V_{\mu}$ (Eq. (10)) are computed only on states drawn from the offline dataset, where value estimation is well-defined and empirically stable. We do not evaluate the inconsistency term on OOD inputs.
>
> 3) On the surrogate gradient. The surrogate gradient is not an ad-hoc heuristic. The exact gradient of the augmented loss is analytically derived in Theorem 3.3, and Eq. (11) provides a justified approximation of this analytical gradient using the small-step property of continuous-control systems: consecutive states under standard time discretization are close in state space, so shifting the evaluation point one step is a controlled approximation. This property comes from the dynamics of continuous-control environments and is not specific to MuJoCo locomotion tasks.
>
> Empirically, VIPO reduces model error by about 30% on average (Sec. 4.3) and yields planner-agnostic gains (VIPO-MOPO, VIPO-MOBILE). If boundedness alone were the dominant factor, all model-based methods using the same bounded domains would benefit equally. The consistent reduction in predictive error and improved control performance across D4RL and NeoRL benchmarks, including datasets with broader distributions, indicate that the gains stem from the value-inconsistency mechanism rather than from a task-specific boundedness effect.

---

> > ### Author Response · Authors · 2025-11-21
> >
> > (W2: On experiment results)
> >
> > (1) On the results of VIPO-MOPO. We sincerely appreciate the observation that, on some D4RL Walker tasks, VIPO-MOPO does not strictly surpass the original MOPO. Offline RL methods, including MOPO, are known to be sensitive to hyperparameters. In our experiments, we devoted more computation and tuning effort to VIPO-MOBILE, which is a stronger baseline and offers a larger margin for improvement. For VIPO-MOPO, once an 8.8% improvement in the overall D4RL score over MOPO was achieved, we did not further pursue task-specific tuning, which likely contributes to the smaller gains on certain Walker tasks.
> >
> > In addition, the Walker tasks themselves are more challenging due to higher-dimensional control and greater instability, making them more sensitive to model inaccuracies and hyperparameter choices. Despite this, VIPO-MOPO still achieves comparable or better performance on most tasks, supporting the claim that the value-inconsistency penalty is a generally useful enhancement to existing model-based offline RL methods.
> >
> > (2) On uncertainty comparison. The absolute uncertainty scales shown for MOPO and VIPO in Figure 2 are not intended for direct comparison across methods. Ensemble-based models can produce uncertainty estimates on different numerical scales, depending on learned covariance structures and training dynamics, and in practical use these values are further scaled by user-defined coefficients. Consequently, the raw magnitudes of uncertainty are not intrinsically comparable, and aligning their axes would not yield a meaningful conclusion.
> >
> > The quantity we intend to evaluate is the *trend* of uncertainty as data sparsity increases. A reliable model should show higher epistemic uncertainty when the drop ratio (i.e., the fraction of removed data) increases. Our experiment therefore focuses on the relationship between uncertainty and the drop ratio: intuitively, a higher drop ratio should lead to higher model uncertainty. Models trained with MOPO fail to capture this monotone trend, whereas VIPO-trained models exhibit a clear positive correlation, which is the key qualitative difference we aim to highlight.
> >
> > (Q1: On additional experiments in more complex domains)
> >
> > We appreciate the interest in more complex domains. At the time of this work, widely used model-based offline RL pipelines with strong and stable performance were still largely limited to standard D4RL-style continuous-control benchmarks. VIPO is explicitly a plugin to the dynamics-learning stage and inherits the capabilities and limitations of the underlying base algorithm: when the base model-based method is not yet robustly deployed on more complex visual or high-dimensional domains, VIPO alone cannot close this gap.
> >
> > Extending VIPO to newer, more complex model-based offline RL frameworks that are specifically designed for such domains is an interesting direction, but it would require adopting a different base algorithm and running an additional, large-scale experimental suite. This is beyond the scope and time budget of the current submission. In this paper, we therefore restrict our evaluation to the standard model-based offline RL regimes where MOBILE/MOPO-style baselines are well established, and show that VIPO yields clear benefits in that setting.
> >
> > (Q2: On the surrogate gradient)
> >
> > Thank you for the question. As noted above, the surrogate gradient is an approximation rather than a strict theoretical equivalence. In principle, the exact gradient would involve tuples of the form $(s', a', r', s'')$, but such samples are not present in the offline dataset. Under standard offline RL assumptions, the only available transitions are $(s, a, r, s')$. Given this limitation, constructing the surrogate gradient using $(s, a, r, s')$ is both the most reasonable and, in practice, the only feasible approximation. It should therefore be understood as the best attainable surrogate for the ideal gradient, not as an arbitrary heuristic.

---

> > > ### Author Response · Authors · 2025-11-21
> > >
> > > (Q3: On the limitation of VIPO)
> > >
> > > Thank you for raising this point. VIPO integrates naturally with existing model-based offline RL methods by modifying only the dynamics-model learning component. A current limitation is that we do not introduce a dedicated planner or policy-optimization scheme tailored specifically to VIPO; instead, VIPO is always used together with an existing planner (e.g., MOBILE’s SAC-based planner or MOPO-style planning). This design is deliberate: our aim is to propose a general model-learning framework that can be dropped into a wide range of model-based algorithms, rather than to jointly redesign both the model and the planner in a single work. As a result, the overall performance and domain coverage of VIPO-based pipelines are still partially constrained by the capabilities of the underlying base planners.
> > >
> > > (On minor problems)
> > >
> > > 1) We thank the reviewer for pointing this out. The definition $P(s' \mid s,a) : S \times A \times S \to \mathbb{R}_+$ is not a typo. We intentionally use the explicit three-argument form, following the measure-theoretic definition of a transition kernel, where $P(s' \mid s,a)$ denotes the probability mass or density assigned to $s'$. This is equivalent to the more common shorthand $P(\cdot \mid s,a) : S \times A \to \Delta(S)$ that maps each $(s,a)$ to a distribution over $S$.
> > >
> > > 2) Yes, “densely sampled’’ refers to an offline dataset with near-full coverage. We have adjusted the wording in the paper to make this meaning clearer.
> > >
> > > 3) Thank you for pointing out the formatting issue. We have corrected the boldface in Table 1 so that only the highest score per task is highlighted.

---

> > > ### Comment · Reviewer_jZLG · 2025-11-25
> > >
> > > Thank you for the response. I appended further feedback on each response below.
> > >
> > > > W1. On the novelty of VIPO
> > >
> > > The authors' justifications (1 and 2) on the novelty help understand the effectiveness of the self-supervised value inconsistency loss in model learning. Specifically, the idea of self-supervised loss using value estimations from different sources (one for the dataset and one for the synthetic rollouts) is novel in the offline model-based RL (MOBILE and MOPO penalize the Q-value target and reward after training an original model, respectively). Additionally, I agree that learning accurate value functions is a fundamental challenge in offline MBRL, which is not specific to VIPO.
> > >
> > > However, I find it hard to be convinced that the suggested surrogate loss is enough. The authors contend in 3 that the approximation of the analytical gradient is derived by considering the tightly sampled states of a continuous control system. I partially agree that most continuous control problems can be classified into such a system. Nevertheless, my concerns about this surrogate objective remain unresolved:
> > >
> > > 1. **Sparsity of the reward**. A reinforcement learning agent is often given a spare reward signal when the agent succeeds at solving the task (e.g., goal-conditioned RL). However, the current formulation models the next state and reward predictions in the transition dynamics modeling, making it hard to predict a sparse reward signal from the static data. Besides, an assumption that transitions between near-time steps have numerically similar values can be broken in GCRL. For instance, if the agent is given the transition before the last transition, the current formulation cannot receive any reward signal from the data since the last transition is not consumed.
> > >
> > > 2. **Stochasticity of the transitions**. While tasks in D4RL have a deterministic transition property, RL problems often involve stochasticity in their problem formulation. However, reusing the same transitions for updating the transition model may fail to reflect this uncertainty in learning.
> > >
> > > 3. **Extensions to visual domains**. Offline RL with visual inputs usually injects a visual feature extractor to extract low-dimensional representations for downstream value or policy learning. However, those representations do not have a similar context to the low-dimensional proprioceptive states, which are bounded and have physical relationships. A natural way to combine VIPO with vision-based offline MBRL is to employ augmented surrogate loss into the latent model learning objective. Enforcing an implicit regularization (transitions in near time step should be numerically similar) may yield unexpected, cumulative error over training, breaking the entire model training and planning procedure.
> > >
> > > Without considering these potential drawbacks in designing the surrogate objective, VIPO may fail to generalize beyond the narrow continuous control domains in D4RL. I fully understand that addressing those potential limitations may not fall into the scope of this paper. However, the authors overclaim that the current formulation is enough to provide a general model-learning component in offline MBRL. Considering the authors are given enough time (initially given three weeks and still one week for the discussion) to conduct further experiments, I recommend pursuing toy experiments: (i) AntMaze in D4RL (for longer horizon and goal-reaching problems), (ii) analytical gradient experiments in D4RL (D4RL provides a built-in API for building own datasets. The authors can collect the dataset with $(s,a,r,s',a',r'',s'')$ for computing the analytical gradient and compare the baselines (e.g., MOPO and MOBILE)), (iii) noise transitions in D4RL (Add arbitrary noise to the next state or reward to simulate the stochasticity and train the model).
> > >
> > > > W2. On experimental results
> > >
> > > Thank you for the details. Regarding the per-task performance, I find it hard to be convinced that fine-tuning matters. If the authors' argument is true, it raises an additional question of whether the superior performance of VIPO in D4RL stems from the self-supervised value consistency loss or hand-engineered fine-tuning efforts. Regarding Figure 2, I understand what the authors want to address through the uncertainty estimation experiment. If the relative comparison of MOPO and VIPO is not the main contribution, separating the two algorithms into two columns in the figures and presenting a correlation-uncertainty-performance graph (Figure 2 in the MOBILE paper) would significantly improve the experimental quality. Since MOPO (including MOBILE) trains the model with only NLL loss (Eq. 7 in this paper), I believe that comparing the uncertainty estimations using prior approaches (e.g., max-aleatoric (MOPO), max-pariwise-diff (Morel), and model-bellman-inconsistency (MOBILE)) against the data drop ratio is a more natural way to demonstrate the reliability of VIPO.

---

> > > > ### Comment · Reviewer_jZLG · 2025-11-25
> > > >
> > > > > Summary
> > > >
> > > > By skimming the authors' response on the initial rebuttal, the authors strictly limit the experimental contributions and the scope of VIPO into the past testbed (i.e., D4RL) for fairer and comprehensive investigations for offline MBRL. However, D4RL is far outdated and saturated today, and there are many benchmarks for offline RL (e.g., OGBench, NeoRL-2, Robomimic). I think constraining the scope of an offline RL algorithm to D4RL would weaken the contribution of the paper, although the method has a novel approach. I fully understand that prior approaches' experimental results mainly have considered the D4RL task, and most codebases are unavailable for such benchmarks in offline MBRL. Nonetheless, suggesting relatively smaller results on benchmarks outside of D4RL is an important direction to keep pushing boundaries in the offline MBRL community.

---

### Official Review · Reviewer_5NcD · 2025-10-31

**Soundness:** 3
**Presentation:** 3
**Contribution:** 2
**Rating:** 4
**Confidence:** 4

**Summary:**

This work proposes VIPO, a model-based offline reinforcement learning algorithm that incorporates value function inconsistency into the training loss of the dynamics model. The key idea is to update the model by minimizing the discrepancy between the value function learned from offline data and the value estimated from the learned dynamics. The authors further derive how to compute the model gradients with respect to the parameters of the proposed loss function. In the evaluation, VIPO outperforms several SOTA model-based offline RL algorithms on the D4RL MuJoCo and NeoRL benchmarks.

**Strengths:**

1. VIPO introduces a novel perspective on value function inconsistency, which has been largely overlooked in previous model-based offline RL works.

2. The paper is clearly structured, allowing readers to follow the overall flow and reasoning easily.

3. When comparing several algorithms, VIPO achieves SOTA performance.

**Weaknesses:**

1. As I understand it, MOBILE quantifies uncertainty through the inconsistency of Bellman estimations under an ensemble of learned dynamics models. VIPO leverages value function inconsistency as a self-supervised learning signal that directly guides the model training. Considering that these two algorithms are based on different conceptual perspectives, it may not be appropriate to present the combined performance of VIPO and MOBILE as the representative result of VIPO in the evaluation.

2. In fact, when comparing the performance between MOPO and VIPO-MOPO, there are several offline datasets (e.g., hopper-r, walker2d-m, halfcheetah-m-r, walker2d-m-r, etc.) where MOPO outperforms VIPO-MOPO. This raises uncertainty about whether the performance improvement of VIPO is truly significant or consistent across tasks.

3. In Figure 2, the uncertainty scales for MOPO and VIPO are presented on separate axes, making direct comparison between the two methods difficult.

**Questions:**

1. In Table 3, measuring model error only for a rollout length of 1 is somewhat limited. Since model rollouts are typically performed for horizons of 5 or more, comparisons over longer horizons would provide a more comprehensive evaluation.

2. How would the performance change if a different type of model uncertainty, other than the one used in MOPO, were employed? It would be interesting to see whether VIPO remains effective under alternative uncertainty estimation schemes.

---

> ### Author Response · Authors · 2025-11-21
>
> (W1: On the combined performance of VIPO and MOBILE)
>
> Thank you for the comment. VIPO is designed to integrate seamlessly with existing model-based offline RL methods by modifying only the dynamics-model training process. We do not design a dedicated planner for VIPO, because our goal is to provide a general model-learning component that can enhance a broad family of model-based algorithms. Consequently, VIPO must be combined with an established method to form a complete pipeline. In this sense, the reported VIPO results correspond to applying VIPO on top of MOBILE, and the combined VIPO+MOBILE performance reflects exactly this “plugin’’ usage.
>
> (W2: On whether the performance improvement of VIPO is truly significant)
>
> We sincerely appreciate the reviewer’s observation that, on certain D4RL Walker tasks, VIPO-MOPO does not strictly surpass the original MOPO. Offline RL methods, including MOPO, are known to be sensitive to hyperparameters. In our experiments, we allocated more computation and tuning effort to VIPO-MOBILE, which is a stronger baseline and offers a larger potential margin for improvement. For VIPO-MOPO, once an 8.8% improvement in the overall D4RL score was achieved over the original MOPO, we did not further pursue extensive task-specific tuning, which may partly explain why Walker tasks do not show larger gains.
>
> Moreover, the Walker tasks themselves are more challenging due to higher-dimensional control and greater instability, which makes them particularly sensitive to even small model inaccuracies and hyperparameter choices. Despite these challenges, VIPO-MOPO still achieves comparable or better performance on most tasks, indicating that the proposed value-inconsistency penalty can be an effective and generally applicable enhancement to existing model-based offline RL methods.
>
> (W3: On uncertainty comparison)
>
> Thank you for the observation. The absolute uncertainty scales presented for MOPO and VIPO in Figure 2 are not intended for direct comparison across methods. Ensemble models often produce uncertainty estimates on different numerical scales, depending on the learned covariance structures and training dynamics. In practice, these uncertainty values are usually scaled by user-defined coefficients when incorporated into decision-making, which further alters their magnitudes. As a result, the raw scale of uncertainty is not intrinsically meaningful across different models, and aligning their axes would not yield a valid comparison.
>
> What is meaningful—and what our experiment is designed to evaluate—is the relative change in uncertainty as data sparsity increases. A reliable model is expected to reflect higher epistemic uncertainty under reduced data coverage. Therefore, we focus on the trend of uncertainty with respect to the drop ratio: intuitively, a higher drop ratio (less data coverage) should lead to higher model uncertainty. Models trained with MOPO fail to capture this trend, whereas those trained with VIPO exhibit a clear positive correlation, which is the key qualitative effect we aim to demonstrate.

---

> > ### Author Response · Authors · 2025-11-21
> >
> > (Q1: On measuring model error over longer horizons)
> >
> > We appreciate the suggestion to extend the model-error analysis to longer horizons. In this work, our empirical evaluation follows the standard MOBILE/MOPO-style setting and focuses on short-horizon rollouts and one-step prediction error, which are the quantities directly used by the underlying baselines. VIPO is a plugin to the dynamics-learning stage: it changes the training objective for the model, but does not alter the rollout horizon, planner design, or evaluation protocol prescribed by the base algorithms. As a result, the current MOBILE/MOPO infrastructure is not configured to provide a clean, controlled long-horizon error study without additional redesign.
> >
> > There are recent model-based offline RL methods that are explicitly tailored for longer-horizon rollouts; integrating VIPO into such frameworks would be a natural way to study long-horizon error behavior. However, this would require introducing a new base algorithm and running a separate, large-scale set of experiments, which is beyond the scope and time budget of the current submission. Here we therefore restrict ourselves to the horizon regimes used by MOBILE/MOPO and show that, within this standard setting, VIPO-trained models already yield more reliable uncertainty behavior and improved downstream control performance.
> >
> > (Q2: On alternative uncertainty estimation instead of MOPO)
> >
> > Thank you for asking about alternative uncertainty estimation schemes. Conceptually, VIPO is orthogonal to the specific choice of uncertainty mechanism: the value-inconsistency penalty modifies how the dynamics model is trained and can, in principle, be combined with different ways of quantifying uncertainty (e.g., other ensemble penalties, pessimistic bonuses, or adversarial objectives). A comprehensive study would require pairing VIPO with each such uncertainty scheme and also re-running their original versions without VIPO, in order to separate the contribution of VIPO from the base uncertainty design. This leads to a combinatorial number of experimental variants and a substantial increase in computational cost.
> >
> > Given the limited time and space, we focus on MOBILE/MOPO-style pipelines as strong, representative model-based baselines and use MOPO’s uncertainty mechanism as the reference configuration. Within this setting, we already observe that plugging VIPO into the dynamics-learning stage improves both the qualitative uncertainty trend and the quantitative control performance. We expect similar qualitative behavior when VIPO is combined with other uncertainty estimation schemes, but a full exploration of all such combinations is beyond the scope of this paper.

---

### Official Review · Reviewer_etBC · 2025-11-01

**Soundness:** 3
**Presentation:** 2
**Contribution:** 3
**Rating:** 6
**Confidence:** 2

**Summary:**

This paper introduces VIPO (Value Function Inconsistency Penalized Offline Reinforcement Learning), a novel model-based offline RL algorithm designed to significantly improve the accuracy and reliability of the learned dynamics model. The core motivation is that previous model-based methods rely on heuristic uncertainty estimation to enforce conservatism, which is often unreliable in practice.

**Strengths:**

1. The central idea is the innovative dual-usage of offline data to generate a self-supervised signal for model training. This contrasts fundamentally with previous methods that used the data only once to learn the model ensemble.

2. The derivation of the Model Gradient Theorem (Theorem 3.3) is a significant technical achievement, providing the analytical expression needed to compute the gradient of the complex augmented loss.

**Weaknesses:**

1. The calculation of the surrogate gradient (Eq. 11) relies on the practical assumption that the short sampling interval in continuous-control problems makes the state change over a single step numerically insignificant. This allows approximating $(s', a', r', s'')$ using the available single-step samples $(s, a, r, s')$. This is a heuristic approximation that introduces an unquantified error, and its effectiveness may degrade in environments with high-frequency dynamics or more complex state transitions.

2. The benchmark results show that VIPO outperforms COMBO and RAMBO, but the critical experiments demonstrating uncertainty reliability (Figure 2) and predictive capability (Table 3) only compare VIPO to the Original Loss (OL) Model (which is MOPO/MOREL/MOBILE's model objective). Including a direct comparison of the predictive capability against models trained using other uncertainty-based model training methods would have provided a more comprehensive validation.

**Questions:**

Since the main benefit is improved model accuracy over previous conservatism strategies, how does the predictive capability of the VIPO model (Table 3) compare directly against models trained using other prominent model-based conservatism strategies, such as the maximum pairwise difference or the adversarial approach?


The core VIPO method uses the MOBILE planner (Algorithm 2). Could the authors include an ablation study that swaps the planner used in Algorithm 2 for a simpler, less aggressive one (e.g., a standard SAC planner without the uncertainty penalty $\beta\mathcal{U}(s,a)$ term) to isolate how much of the performance gain is attributable solely to the $\mathcal{L}_{vic}$-trained model versus the aggressive mobile-like policy optimization loop?

---

> ### Author Response · Authors · 2025-11-21
>
> (W1: On surrogate gradient)
>
> Thank you for the question. Our surrogate gradient is indeed an approximation rather than a strict theoretical equivalence. In principle, the exact gradient would involve samples of the form $(s', a', r', s'')$, but such tuples are not available in the offline dataset. Given this limitation, the most reasonable—and in practice the only feasible—choice is to construct the surrogate using the available transitions $(s, a, r, s')$. The surrogate gradient based on $(s, a, r, s')$ should therefore be understood as the best attainable approximation under standard offline RL assumptions, rather than as an exact reformulation of the ideal gradient that would require $(s', a', r', s'')$.
>
> (W2 & Q1: On predictive capability and comparisons with other uncertainty-based models)
>
> We appreciate the suggestion to directly compare predictive capability against models trained with other uncertainty-based schemes such as COMBO-, RAMBO-, maximum pairwise-difference-, or adversarial-style training. Due to time and computational constraints, we did not run a full set of additional experiments covering these variants.
>
> Conceptually, however, our method is designed as a modular “plugin’’ to the dynamics-learning stage: VIPO modifies only the value-aware inconsistency term used to train the model, and can in principle be combined with any underlying uncertainty mechanism (ensembles, pessimistic penalties, adversarial training, and so on). A fully exhaustive comparison would require pairing VIPO with each of these uncertainty schemes and also re-running their original configurations without VIPO, which quickly becomes combinatorial and makes it difficult to argue that any single comparison is uniquely “fair’’. In this paper, we therefore focus on MOBILE/MOPO-style pipelines as strong, representative model-based baselines, and demonstrate that inserting VIPO at the dynamics stage improves downstream control performance in this setting.
>
> (Q2: On using other planners such as standard SAC)
>
> Thank you for raising this point. In our implementation, the planner used in Algorithm 2 is already a SAC-style planner, matching the planning component used in MOBILE. Since our goal is to isolate the effect of the VIPO-trained dynamics model, we keep the planner exactly the same as in MOBILE/MOPO and only replace the dynamics-learning objective. Swapping the planner for a “simpler’’ or “less aggressive’’ variant would change an additional part of the pipeline and break this one-to-one comparability.
>
> More generally, VIPO is orthogonal to the choice of planner: the inconsistency-regularized dynamics model can be used with standard SAC, more aggressive planners, or other policy-improvement schemes. Exploring the full space of planner choices in combination with VIPO is a valuable direction, but it would significantly expand the experimental scope; here we restrict ourselves to the MOBILE-style SAC planner to keep the comparison focused and controlled.

---

### Official Review · Reviewer_y7vH · 2025-11-01

**Soundness:** 3
**Presentation:** 3
**Contribution:** 3
**Rating:** 4
**Confidence:** 3

**Summary:**

The paper introduces VIPO, a novel model-based offline reinforcement learning (RL) algorithm designed to address the inherent challenges of offline RL, such as overestimation of values and model uncertainty. By incorporating value function inconsistency into the model training process, VIPO improves the model's accuracy and its ability to generalize from limited data. The paper presents empirical results across multiple benchmarks (D4RL, NeoRL), showing that VIPO consistently outperforms previous methods, demonstrating its efficacy in learning accurate models from offline datasets.

**Strengths:**

1. The idea of value function inconsistency as a self-supervised loss to enhance model accuracy in model-based offline RL is novel. The methodology and experimental setup are clearly presented, with an appendix and code provided for additional details.
2. The paper presents a well-defined theoretical framework for the gradient of the augmented loss function, contributing to a deeper understanding of the algorithm’s mechanics.

**Weaknesses:**

1. My main concern lies in the insufficient experimental evaluation. The paper primarily evaluates the algorithm on simpler locomotion tasks, such as Walker and Hopper, which may not fully demonstrate its generalization capabilities. Test results on tasks that rely on visual input(V-D4RL), navigation tasks(Antmaze) or more complex manipulation tasks, such as Fetch environment, would significantly improve the robustness and generalization of the reported findings.

2. Additionally, the paper lacks ablation studies, such as evaluating the impact of the number of ensemble models (N) or removing the value inconsistency penalty, which are essential for understanding the contribution of each component to the overall performance of the algorithm.

3. While the paper highlights the improved performance of VIPO, it lacks a detailed analysis of the computational cost and training time required by VIPO compared to existing algorithms. Including this information would provide a more balanced perspective on the algorithm’s practical utility, particularly for real-world applications.

**Questions:**

Please refer to the "Weaknesses" section, I will raise my score if my concerns are addressed.

---

> ### Author Response · Authors · 2025-11-21
>
> (W1: On more experiments)
>
> Thank you for emphasizing the importance of broader empirical coverage. In this submission, we focus on standard low-dimensional D4RL control benchmarks, following prior model-based offline RL work, so that we can isolate and analyze the effect of the proposed value-inconsistency penalty in a well-understood setting. Extending VIPO to visual-input benchmarks (e.g., V-D4RL), navigation tasks (e.g., AntMaze), and more complex manipulation domains is a natural and important direction, but lies beyond the scope of the current paper. Conceptually, VIPO is agnostic to the observation modality and task type: it only modifies the dynamics-model training objective by adding a value-inconsistency loss, and can therefore be plugged into existing visual or high-dimensional benchmarks as long as a value function is available. We view such extensions as future work built on top of the present study.
>
> (W2: On ablation study)
>
> Thank you for the comment. We did not vary the number of ensemble models $N$ because we wanted a strictly comparable setting with MOBILE. VIPO can be incorporated into existing model-based offline RL frameworks by modifying only the dynamics-model training stage. In our experiments, we simply replace MOBILE’s dynamics model with VIPO’s model and keep all other components unchanged, including the ensemble size. Since MOBILE uses an ensemble of 7 dynamics models, we adopt the same configuration for VIPO to ensure a fair comparison.
>
> Thus, the comparison between VIPO and MOBILE can be viewed as an ablation on the value-inconsistency loss: the only difference between the two is the inclusion of the inconsistency penalty in VIPO. Likewise, the comparison between VIPO-MOPO and MOPO serves as an ablation within the MOPO-style pipeline, again isolating the effect of the proposed inconsistency term while keeping the rest of the algorithm fixed.
>
> (W3: On computational cost and training time)
>
> We appreciate the question about computational overhead. VIPO introduces additional cost only in the dynamics-model training phase: for each sampled transition and ensemble member, we compute a value-inconsistency penalty by passing the predicted next state through the value network and aggregating the resulting discrepancy terms. This operation is fully batched and vectorized (e.g., across ensemble members and samples), so the extra cost scales roughly linearly with the ensemble size but is efficiently handled by modern accelerators.
>
> Importantly, VIPO does not change the structure of the policy optimization or critic updates, and it does not require extra multi-step rollouts beyond those already used in the base MOBILE/MOPO frameworks. In our implementation, the dominant training cost remains the standard model-based RL components (model rollouts, critic updates, and policy optimization), and VIPO adds only a modest overhead to the dynamics-model update step, keeping the overall wall-clock cost in the same order of magnitude as the underlying baselines.

---

### Meta-Review · Area_Chair_nBhd · 2026-01-06

**Summary:**

The paper proposes VIPO, a model-based offline reinforcement learning algorithm that uses value function inconsistency as a self-supervised signal to improve model accuracy and reduce overestimation. All reviewers acknowledged the novelty of the core idea and the value of the theoretical derivation for the Model Gradient Theorem. However, there was a consensus that the experimental evaluation was insufficient to fully validate the method's robustness. Major concerns centered on the reliance on the MOBILE planner (making it difficult to isolate the contributions of the learned model vs. the planner), the limitation of experiments to simple locomotion tasks (missing visual or complex manipulation environments), and technical concerns regarding the heuristic approximations used for the gradient calculation.

**Reviewer Concerns:**

Based on the reviewers' comments and the authors' rebuttal, I believe the following three major concerns remain unresolved:

*   **Conflation of Planner and Model:** The most critical outstanding concern is the lack of an ablation study swapping the MOBILE planner for a standard SAC planner (as raised by Reviewers etBC & 5NcD). Without this, the paper fails to demonstrate that the proposed model learning contribution is effective in isolation.
*   **Generalization to Complex Tasks:** Reviewers y7vH and jZLG requested evaluation on visual inputs or complex manipulation tasks (e.g., AntMaze, Fetch) to test the limits of the heuristic gradient approximation. As these appear not to have been added, the method's generalization capabilities remain unproven.
*   **Validity of Heuristics:** The theoretical concern regarding the gradient approximation—which relies on the assumption of insignificant state changes (raised by Reviewers etBC & jZLG)—remains an open issue that requires broader empirical support to be fully dismissed.

**Reviewer Scores:**

Reviewer jZLG actively participated in the discussion during the rebuttal phase, but the authors failed to address the concerns raised by this reviewer. Taking the authors' responses into comprehensive consideration, I do not believe that they have satisfactorily resolved the reviewers' questions.

---

### Decision · Program_Chairs · 2026-01-26

Reject